


# A Dynamic Bidirectional Coupled Hydrologic-Hydrodynamic Model for Flood Prediction

Chunbo Jiang[1], Qi Zhou[1], Wangyang Yu[1,2] , Chen Yang[3] and Binliang Lin[1]

[1]State Key Laboratory of Hydroscience and Engineering, Department of Hydraulic Engineering, Tsinghua University, Beijing, 100084, China
[2]China Institute of Water Resources and Hydropower Research, Beijing, 100038, China
[3]School of Water Conservancy, North China University of Water Resources and Electric Power, Zhengzhou, 450046, China

*Correspondence to*: Wangyang Yu (sea198719@126.com)

**Abstract.** As one of the main natural disasters, flood disaster poses a great threat to township development and property security. Numerous hydrological models and hydrodynamic models have been developed and implemented for flood simulation, risk prediction and inundation assessment. In this study, a dynamic and bidirectional coupled hydrodynamic-hydrologic-hydrodynamic model (DBCM) is developed to predict and evaluate inundation impact in a catchment in mountain area. Based on characteristic theory, the proposed method is able to dynamically adapt and alternate the simulation domain of hydrologic model, and/or hydrodynamic model according to the local flow condition, and a key feature of the proposed model is the dynamic coupling splitting the hydrologic and hydrodynamic simulation domains. The proposed model shows good prediction accuracy and overcomes the shortage existing in previous unidirectional coupling model (UCM). Existing numerical examples and physical experiments were both used to validate the performance of DBCM. Compared to UCM, results from DBCM show good agreements with analytical and measured data which indicates that the proposed model effectively reproduces flood propagation process and accounts for surface flow interaction between non-inundation region and inundation region. Finally, DBCM is applied to predict the flood in the Longxi river basin, and the simulation results show the capability of DBCM in conducting flood event simulation in interested catchment which can support flood risk early warning and future management.

## 1 Introduction

Over the past decades, flood events occurred frequently as one of the most devastating natural hazards which impact millions of people across the world, as a result of global warming, population growth, rapid urbanization and climate change (Zhu et al., 2016). Between 1998 and 2016, economic loss due to flood induced disasters has reached millions of yuan in China (Osti, 2017). Thus, prediction and early warning of flood events plays an important role in the flood risk assessment and management as well as urban design and policy-making.

With the advances in computation and information technology, a large number of studies have been carried out to investigate the hydrologic process and assess flood risk. Numerous hydrologic models and hydrodynamic models have been proposed to



deal with these related problems (Li et al., 2016; Leandro et al., 2014; Li et al., 2013; Singh et al., 2015; Yu and Duan, 2014; Yu and Duan, 2014). The purpose of the hydrologic model place emphasis on accurate simulating the temporal processes and responses of water cycle between and within atmosphere, surface and soil over a wide range of space and time scales. Both of lumped and distributed hydrologic model are commonly used to conduct hydrologic processes simulation(Singh and Woolhiser,

2002), and the outputs of most hydrological models only cover time-dependent discharge at controlled outlets without a full description of flow information across the entire domain. Whereas, hydrodynamic models solve physics based mathematical equations to conduct simulation of the interested catchment over two-dimensional computational domains, and full information of the flow in the study domain can be obtained (Yu and Duan, 2014; Patro et al., 2009; Singh et al., 2015), such as the water depth, flow velocity, flow duration, etc. However, as a prerequisite for starting a hydrodynamic model simulation, specific

data on boundary conditions need to be prepared from other sources, such as hydrologic model results, historical records and real-time monitoring data, etc. Therefore, it has been a hot topic for decades in flood prediction research that effectively coupling hydrologic model with hydrodynamic model so as to solve the problem stated above and making the full use of their own characteristics of both models.

At present, a number of hydrological-hydrodynamic coupled models have been proposed and developed for flood

assessment. These models can be classified into two main categories: unidirectional coupling model (UCM) (Montanari et al., 2009; Choi and Mantilla, 2015) and bidirectional coupling model(BCM) (Zhu et al., 2016; Thompson, 2004). In terms of unidirectional model which is the most widely implemented method in real practice, the hydrologic sub-model is performed at the first stage to obtain hydrographs, and the data obtained feed the hydrodynamic sub-model subsequently. Thus, for UCM, the flow information is transferred from the hydrologic model to the hydrodynamic model in one direction without water

exchange between these two sub-models. Nevertheless, the UCM has the advantages of easy to run and making full use of existing models without the need for users to modify or rebuild the original models. (McMillan and Brasington, 2008) developed a coupled precipitation-runoff hydrological model with 1D dynamic wave model being used to assess the flood inundation for several flood return periods. Other researchers (Choi and Mantilla, 2015; Grimaldi et al., 2013; Montanari et al., 2009) adopted similar coupling methods to investigate flood risks. Many advanced opens-source and commercial modelling

packages (SWAT(Liu et al., 2015), HEC-HMS(Hdeib et al., 2018), DHI MIKE(Rayburg and Thoms, 2009), etc.), can be easily applied to UCM.

Although UCM is easy to use, it is unable to describe the natural flow processes. The output results from the hydrological model are taken as the inflow boundary condition for the hydrodynamic model, and the boundary conditions are fixed throughout the computation. They may introduce errors because the surface runoff yield in the hydrodynamic domain

hydrodynamic is not involved in the flood simulation (Lerat et al., 2012) and overestimate the flood risk in some extent. However, for a real flood event, the process of rainfall-runoff production can occur at any location within the study basin, and the inflow position is time varying. One solution to solve this problem is to use rainfall datasets as input data, and then employ BCM to link the hydrologic model with the hydrodynamic model. In line with this objective, various techniques have been proposed, ranging from simple approach through changing boundary conditions, such as point source or lateral flow conditions



(Bouilloud et al., 2010), to relatively complicate models, such as using the simplified 2-D shallow water equations to simulate overland flow instead of traditional hydrologic model (Viero et al., 2014)(Muskingum method, etc) which only consider the precipitation and infiltration processes. The coupled MIKE SHE/MIKE 11 modelling system (Thompson, 2004; Thompson et al., 2004) is one kind of these BCMs that the hydrodynamic and hydrologic models can exchange discharges at pre-specified reaches, where the flow velocity is computed based on water level gradient between MIKE SHE and MIKE 11, and the

calculated flow be treated as lateral flow when solving the momentum equation of the hydrodynamic model. The next step is to determine whether the discharge term is inflow or outflow fed back to the hydrodynamic model in the next time step. In this method, the current water level of the hydrologic model and hydrodynamic model have been used in velocity calculation at the mutual boundary, It does not consider present flow state (Bravo et al., 2012; Laganier et al., 2014). It is different from BCM using the lateral inflow conditions that velocity provided by hydrologic model will be added to the governing equations

of hydrodynamic model directly, not considering present flow state.

The existing coupling methods, either UCM or BCM,  still have some shortcomings in the simulation of flood propagation processes. On the one hand, the location of the joint boundary between the hydrologic and hydrodynamic models, where these two models exchange flow information, is predetermined. Generally, for a coupling model domain, non-inundation region the hydrological model is used and inundation region the hydrodynamic model is used. However, in a real flood event, the non-

inundation and inundation regions may change according to the predefined flow state. Whereas, a significant problem may occur if boundary position is specified in advance.  A very large inundation region will cost more computing resources and reduce efficiency, while a very small inundation region may lead to the flood area being located beyond the pre-set boundary. Thus, the size of domain is a key issue in coupling models. Secondly, the discharge at the boundaries of the two types of models ought to be calculated by hydrologic model and hydrodynamic model, considering of the water depth and velocity.

The UCM feed hydrodynamic model with the output of hydrologic model as the inflow boundary, and does not assess the feedback effect from the hydrodynamic model, which has been taken into account by the BCM. The existing BCM consider the water volume exchange between two models without precise consideration of local velocity information. Taking MIKE SHE/MIKE 11 coupling model as an example, the grid velocity on coupling boundary is temporal calculated based on flow depth difference between the two models, and then the obtained temporal velocity is used to solve momentum equations in

hydrodynamic model before determining whether the flow impact from hydrologic model is side inflow or outflow. This approach is apt to conduct and perform, while the temporal velocity still doesn't take their own original velocity of both models into consideration which limits its application only to 1D flow. Thus, further study is necessary to be done for more general implementation, such as 2D flow or other more complicate cases. In order to reach the goal of dynamic coupling and keep mass and momentum conservation, the flow states from both hydrologic and hydrodynamic models on the coupling boundary

should be taken into consideration which means the grid-self flow depth and velocity cannot be discarded. Besides, special focus should be paid to the boundary dynamic change and subsequent flow states after the determination of discharge variation.

The aim of this study is to develop a dynamic bidirectional coupling hydrologic-hydrodynamic model (DBCM) capable of realizing the dynamic switching of applied hydrologic and hydrodynamic models. A two-dimensional hydrodynamic model



and a rainfall-runoff hydrologic model are coupled based on the techniques of characteristic wave theory. In comparison to
existing approaches, the main advantages of DBCM proposed in this paper are two folds, (1) a dynamic coupling approach of
hydro-hydrodynamic model based on characteristic wave theory is developed for the first time, and the running process of the
model is consistent with the natural flood propagation; (2) the flow calculation on the coupling boundary is the key point based
on the theory of characteristic to realize the dynamic switch of the surface flow simulation within both models,
comprehensively considering the current flow state computed by both models.

The methodology of the proposed DBCM is described in section 2. After that, the performance of the proposed model is
verified by numerical and physical experiments in Section 3, as well as comparison and discussion with former approaches. In
section 4 the DBCM is applied to the Longxi river catchment in Chongqing City, and then followed by conclusions.

## 2 Methodology

The  DBCM model comprise a hydrologic model  which includes three sub-models (rainfall, infiltration and slope runoff) and
a hydrodynamic model which solves 2D shallow water equations used to simulate channel and overland flow. Both models
are solved simultaneously at each time step, and flow information on the coupling boundary is calculated based on the theory
of characteristic wave propagation commonly employed in solving Riemann problems (Toro, 2001).

### 2.1 Hydrologic model

The hydrological model used in this study is a physics, raster-based, and distributed model. The runoff yield of a catchment
involves the processes of precipitation and infiltration. 2-D diffusion wave equations is used in overland flow modelling.

The precipitation module reads in record datasets from a rainfall station and interpolates the data over the whole
computational domain using a spatial interpolation function (Thiessen polygon method, Inverse Distance Weighted, etc.). The
infiltration model solves the Green-Ampt equation (Rawls et al., 1983), a theoretical formulation obtained based on Darcy
formula with a simpler form as follows.

$f_\mathrm{p} = K_s \left( 1 + \frac{(\theta_s - \theta_i)S_a}{F_c} \right)$ ,                                         (1)

where $f_p$ is the infiltration rate(mm h$^{-1}$), $K_s$ is the hydraulic conductivity(mm h$^{-1}$), $S_a$ is the average effective suction of the
wetting front (mm), $\theta_s$ and $\theta_i$ are saturated and initial soil moisture content respectively (%), $F_c$ is the cumulative infiltration
(mm). According to the relationship between infiltration rate, soil moisture content and rainfall intensity, this formula can
reflect runoff yield conditions under whether saturated storage or excess infiltration, and it has been widely verified and works
well.

Surface flow routing models can be divided into conceptual hydrologic model and physical hydrologic model. The
conceptual model, such as Soil Conservation Service(SCS) formulation (Rallison and Miller, 1982) ,an empirical model for
estimating the amounts of runoff under varying land use and soil, and unit line formulation, commonly output the runoff




hydrographs at control section, but it is not capable of providing detailed information about the water movement over the entire

basin. Moreover, the location of the control section and computing grid cannot be changed once determined. The mesh generation principle of the conceptual hydrologic model is not consistent with that of the hydrodynamic model. Therefore, the conceptual hydrologic and hydrodynamic models cannot be processed using the same computational grid model. Hence the conceptual hydrologic model and hydrodynamic model can only be solved sequentially and independently. Nevertheless, the governing equations of the process based hydrologic model often take advantage of the simplified forms of hydrodynamic

model(kinematic wave model (Borah and Bera, 2000), diffusion wave model (Leandro et al., 2014; Downer et al., 2002), etc.) to simulate the flow routing process. A dynamic switch between the process based hydrologic and hydrodynamic models is implementable, as a result of the numerical solution procedure and mesh generation principle are consistent.

The diffusion wave equations (Bates and De Roo, 2000) are used to determine the runoff routing, which is composed of mass conservation equation and momentum equations:

$$\frac{\partial h}{\partial t} + \frac{\partial q_x}{\partial x} + \frac{\partial q_y}{\partial y} = Q_m, \tag{2}$$

$$Q_x = \frac{A_x R_x^{0.67} S_x^{0.5}}{n}, \tag{3}$$

$$Q_y = \frac{A_y R_y^{0.67} S_y^{0.5}}{n}, \tag{4}$$

where $q_x, q_y$ are unit discharges along the $x$ and $y$ directions(m$^2$ s$^{-1}$), $h$ is water depth(m), $Q_m$ equals to rainfall rate minus infiltration rate (m s$^{-1}$), $Q_x, Q_y$ are flow rate in the direction of $x$ and $y$ (m$^3$ s$^{-1}$), respectively, $A$ is flow area (m$^2$), $R$ is

hydraulic radius (m), $S$ is water level gradient, and n is roughness coefficient.

Since the effect of acceleration and inertial terms of water flow on the urban surface is not significant compared to gravitational and frictional terms (Chen et al., 2012; Hsu et al., 2000), the time dependent terms in the original momentum equations are omitted, thus two diffusive wave equations are obtained. The numerical scheme can be found in the JFLOW model (Bradbrook et al., 2004; Yu and Lane, 2006). The diffusive wave model does not compute the flux term in the

momentum equations. Velocity entirely depends on the local water level gradient and roughness, and water depth relates to discharge from the neighbour grid. The possible flow is up to two of the adjacent cells at each time step:

$$Q_i = \frac{w h^{5/3} S_i}{n \left( S_i^2 + S_j^2 \right)^{1/4}}, Q_j = \frac{w h^{5/3} S_j}{n \left( S_i^2 + S_j^2 \right)^{1/4}}, \tag{5}$$

where

$$S_i = \frac{\eta_{i,j} - \eta_{i\pm1,j}}{w}, S_j = \frac{\eta_{i,j} - \eta_{i,j\pm1}}{w},$$

$$h_i = \eta_{i,j} - \max(z_{i,j}, z_{i\pm1,j}), h_j = \eta_{i,j} - \max(z_{i,j}, z_{i,j\pm1}),$$

$$h = \frac{h_i S_i^2 + h_j S_j^2}{S_i^2 + S_j^2}, \tag{6}$$

where $w$ is the width of the cell(m), $S_i, S_j$ are water level slope in the orthogonal direction of $i$ and $j$, respectively, $h_i, h_j$ are effective depth in orthogonal direction of $i$ and $j$, respectively, $\eta_{i,j}$ and $z_{i,j}$ are the water surface level and ground elevation(m),




respectively, and $h$ is the effective depth. The change of water depth in each of the cells is then calculated using the following

equation:

$$\Delta h = \frac{(\sum Q_{in\,i,j} - \sum Q_{out\,i,j} - Q_m)\Delta t}{w},$$ (7)

### 2.2 Hydrodynamic model

The governing equations for the hydrodynamic model are the widely used 2D shallow water equations. Neglecting the Coriolis

force term, wind resistance term and viscosity term, the equations are composed of the continuity equation

$$\frac{\partial h}{\partial t} + \frac{\partial hu}{\partial x} + \frac{\partial hv}{\partial y} = Q_m,$$ (8)

and the momentum equations

$$\frac{\partial hu}{\partial t} + \frac{\partial}{\partial x}\left(hu^2 + \frac{1}{2}gh^2\right) + \frac{\partial}{\partial y}(huv) = -gh\frac{\partial z}{\partial x} - C^2 u\sqrt{u^2 + v^2},$$ (9)

$$\frac{\partial hv}{\partial t} + \frac{\partial}{\partial x}(huv) + \frac{\partial}{\partial y}\left(hv^2 + \frac{1}{2}gh^2\right) = -gh\frac{\partial z}{\partial y} - C^2 v\sqrt{u^2 + v^2},$$ (10)

where $u, v$ are velocities along the $x$ and $y$ direction(m s$^{-1}$), respectively, $h$ is water depth(m), $g$ is gravity acceleration
(m s$^{-2}$), $z$ is bottom elevation(m), $C$ is Chezy coefficient representing roughness, $Q_m$ is the source term which equals to
rainfall rate minus infiltration rate (m s$^{-1}$).

The finite volume method following TELEMAC (Ata et al., 2013) are used to solve these equations. And the convection
flux on grid faces is calculated using the HLL scheme with WAF approach (Toro, 2001).

$$\begin{cases} F^{hll} = F_L & S_L \geq 0 \\ F^{hll} = \frac{S_R F_L - S_L F_R + S_L S_R(U_R - U_L)}{S_R - S_L} & S_L \leq 0 \leq S_R, \\ F^{hll} = F_R & S_R \leq 0 \end{cases}$$ (11)

$$S_L = U_L - \sqrt{gh_L}, S_R = U_R + \sqrt{gh_R},$$

where $U_L, U_R, h_L, h_R$ are the components of the left and right Riemann states for a local Riemann problem, and $S_L, S_R$ are
estimates of the speeds of the left and right waves. $F^{hll}$ is the fluxes in the middle region. Based on this flux, the WAF method
guarantees a second order accuracy in time and space is proposed:

$$F_{i+\frac{1}{2}} = \sum_{k=1}^{N+1} \beta_k F^{(k)}_{i+\frac{1}{2}}$$

$$\beta_k = \frac{1}{2}(c_k - c_{k-1}), c_k = \frac{\Delta t S_k}{\Delta x}, c_0 = -1 \text{ and } c_{N+1} = 1,$$ (12)

where $F^{(k)}_{i+\frac{1}{2}} = F(U^{(k)})$, $N$ is the number of waves in the solution of the Riemann problem, and $\beta$ corresponds to the

differences between the Courant numbers $c_k$ of successive wave speeds $S_k$.

The topography term on the right side of equation (9) and (10) is calculated by the hydrostatic reconstruction scheme:


$\quad -gh\frac{\partial z}{\partial x} = \nabla\frac{gh^2}{2} = \frac{g}{2}\frac{\Delta t}{\Delta x}[(h_i^R)^2 - (h_i^L)^2]$ , $\qquad\qquad$ (13)

$$\begin{cases} h_i^R = max[0.0, h_i + z_i - max(z_i, z_{i+1})] \\ h_i^L = max[0.0, h_i + z_i - max(z_{i-1}, z_i)] \end{cases} ,$$

The friction term is computed by a semi-implicit scheme to ensure numerical stability (Liang et al., 2007):

$$(hu)^{n+1} = \frac{(hu)^n}{1+\Delta t\left(\frac{g\sqrt{(hu)^2+(hv)^2}}{h^2 C^2}\right)^n} , \qquad\qquad (14)$$

The time step is determined by CFL condition. More details of the numerical schemes can be referred to (Ata et al., 2013).

## 190    2.3 Dynamic bidirectional coupling model(DBCM)

The hydrologic model and hydrodynamic model in DBCM are solved simultaneously. The main features of the DBCM are: (1) the computation domain is divided into a non-inundation region and an inundation region, and the hydrologic model is solved in non-inundation region while the hydrodynamic model is in inundation region. Whether the hydrologic or hydrodynamic model is implemented in a specific grid is determined based on its own and neighbouring flow state, and the

location dividing the non-inundation region and inundation region forms the dynamic coupling boundary which is time dependent; (2) the flow rate calculation on coupling boundary takes full account of the current flow state computed by hydrologic model and hydrodynamic model.

     The hydrologic model is used to calculate the overland flow in a non-inundation region with a small water depth, and the hydrodynamic model is used to simulate the flood propagation process in an inundation region. The model applied to one

location at different times may be changed according to the local water depth fluctuation, and the boundary location where flow enters the inundation region is also changing constantly. As shown in Fig.1, with the increasing of rain intensity, the inundation region expands as a consequence of the gradually accumulating the surface water volume. The positions of the inlet flow boundary, flow path and discharge change subsequently, and vice versa. The coupling models proposed by other researchers, either UCM or BCM, hardly consider this phenomenon.




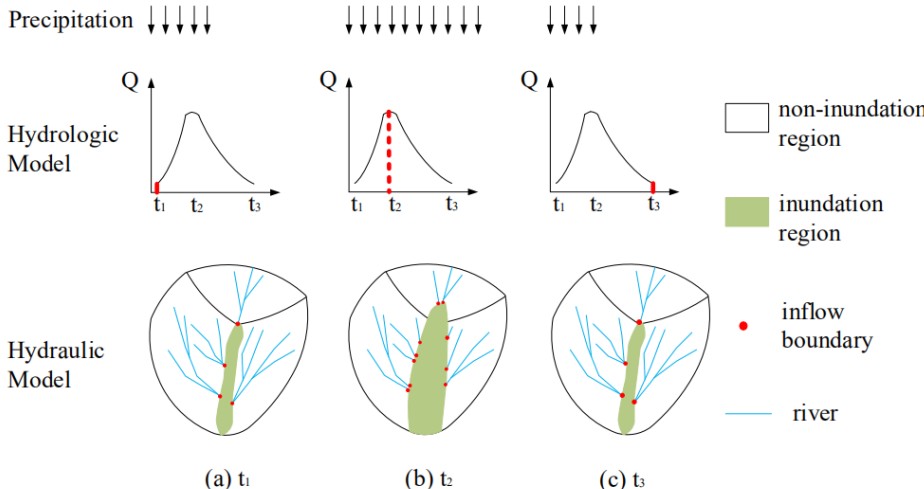

**Figure 1. Schematic diagram of DBCM**

Figure 2 shows a detailed process of flow state change on both sides of the coupling boundary, and the resulting position transition of the coupling boundary between the non-inundation region(zone 1) and inundation region(zone 2). In the case the slope confluence flows to the river, and water flows in the inundation region from the slope to the river, as shown in Fig.2a and Fig.2b, the discharge at the coupling boundary equals to the upstream discharge and not affected by the downstream flow, which means the local discharge is completely determined by the flow routing calculation in the hydrologic model. After the water depth is updated, the location of the coupling boundary is moved to point A based on the water depth threshold, which is defined to distinguish the two regions. Moreover, in the inundation region the flow may move from downs to upstream, as shown in Fig.2c and Fig.2d. The discharge at the coupling boundary may be determined by both upstream and downstream flows. In this case, if the upstream slope flow is assigned directly to the discharge on the coupling boundary, an error will inevitably occur. Therefore, the discharge at the coupling boundary is calculated on the basis of current flow states in zones 1 and 2. Then, the same process will be performed  to update water depth as well as the  new distribution of inundation and non-inundation regions. When the inundation zone expands due to water level rise, the coupling boundary location moves to point B.




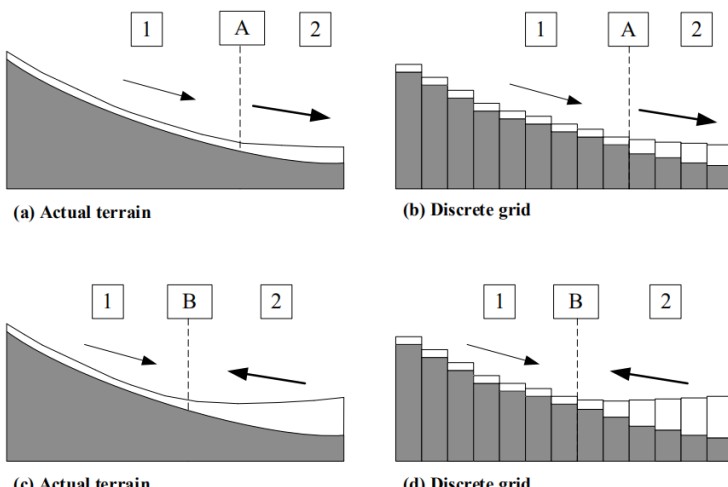

**Figure 2. Flow state change on both sides of the coupling boundary and resulting position of the coupling boundary**

In previous studies, the discharge at the coupling boundary may be computed directly through the hydrologic model, using empirical formulae, or by interpolation according to the water level or velocity gradient on both sides of boundary. Such methods may still fail to provide an overall understanding of the flow regime status of the combined hydrologic and hydrodynamic model. The DBCM is conducted following the procedures at the coupling boundary: the flow state is obtained by both the hydrologic and hydrodynamic models in their local grids, then the discharge through the coupling boundary is computed and the entire water depth is updated according to the water volume variation. After that, the location of the coupling boundary is updated and the relative area of non-inundation region and inundation region are remapped. The key issue of DBCM is how to establish a reasonable approach to compute the discharge on the coupling boundary, which need to integrate the effect of current flow state obtained by the hydrologic and hydrodynamic models on both sides of the coupling boundary.

According to Godunov theory(Godunov, 1959), the solution of a convective flux using the finite volume method is considered as a local Riemann problem. The grid discontinuity characteristic speed represents the propagation of local fluid variables in time and space, as shown in Fig.3. When the characteristic speeds are all positive, the flux depends entirely on the left-side flow state, and vice versa. However, when the characteristic speeds have a negative value and a positive value, both the current flow state in the two grids must be taken into consideration. Applying this theory to DBCM, the computational scheme at the boundary can be specified. It is known that the hydrologic model only transfers water mass, while the hydrodynamic model transfers both water mass and momentum. More details of different coupling cases are shown Fig.4.


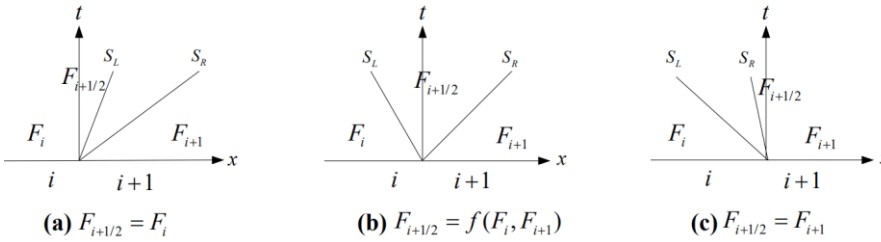

**Figure 3. Direction of Characteristic wave**

For case A, the hydrologic and hydrodynamic models are calculated independently, corresponding to the situation that positive bed slopes inducing confluence flows into the river, thus only the discharge calculated by the hydrological model passes through the coupling boundary (Fig. 2a and 2b). The flow values at grids k and i are calculated using the hydrologic model and at grid j is solved by hydrodynamic model, see Fig. 4. Firstly, slope analysis of diffusion wave equations is applied uniformly. Obviously the water level gradient between k and i is smaller than that of i and j. According to the calculation

results from the diffusion wave equations, the velocity is directed to the maximum water level slope. Therefore, the change of water depth in grid k has nothing to do with the flow state at grid i, and the velocity change at k is analysed by other grids on the left of k. The flow information at grid i and j constitute a local Riemann problem and the characteristic speed is analyzed. The velocity at grid i is obtained from above analysis, and the velocity at grid j is the velocity at current moment. The interface water depths at contact discontinuity are calculated: $h_i^r = h_i + z_i - \max(z_i, z_j)$, $h_i^l = h_j + z_j - \max(z_i, z_j)$. Thus a pair of

characteristic wave at the interface are obtained:

$$S_L = u_i - \sqrt{gh_i^R}, S_R = u_j + \sqrt{gh_j^L} , \qquad (15)$$

When the characteristic speeds $S_R \geq S_L > 0$, the flux calculation depends only on the flow information at grid i, independent of that at grid j. The velocity at grid i is calculated using the diffusion wave equations and only outflow is permitted. In addition to the change of water depth calculated according to the hydrodynamic model at grid j, the water volume transferred from grid

i should also be added. That no convection term in the momentum equation of the hydrological model indicates no momentum transfer at the discontinuity between grid i and j, and the velocity of the two grids does not interact with each other.

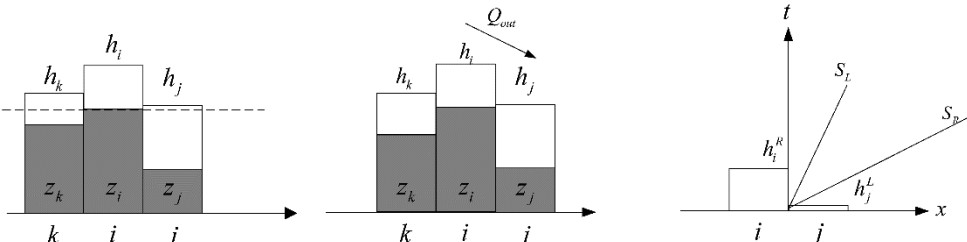

**Figure 4. Coupling condition A: discharge on coupling boundary depends on hydrologic model**

For the second case, the hydrological model and hydrodynamic model are calculated together, corresponding to the situation

that inundation area expands(i.e. Fig.2c and 2d). As shown in Fig. 5, the water depth in grid k and i are small, and the hydrologic





model is applied. While grid j has a deeper water depth and smaller elevation, the hydrodynamic model is applied. In this case, the velocity direction is form grid i to grid k. The characteristic wave analysis at the interface of grid i and grid j reveals that $S_R > 0 > S_L$, which means that the momentum at grid j can be transferred to grid i. Grid i is involved in the computational domain of the hydrodynamic model. The water depth increment at grid i needs to deduct the current discharge output to grid

k and the flow rate obtained by solving the hydrodynamic equation with the flow state at grid j. The velocity increment at grid i is obtained by solving the hydrodynamic equation with the flow state at grid j based on current velocity. Then the flow state at grid i is updated. And coupling boundary position may change when water depth varies.

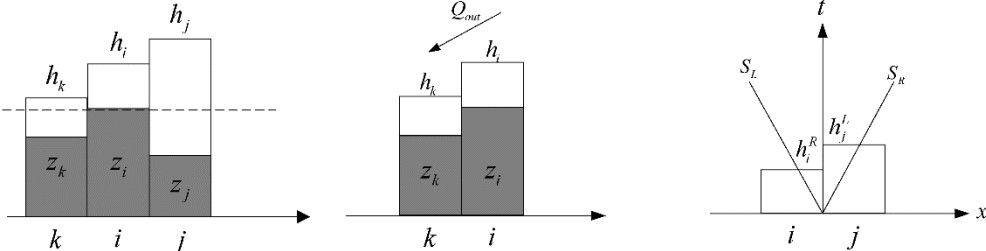

**Figure 5. Coupling condition B: discharge at coupling boundary is determined by both hydrologic model and hydrodynamic model**

The slope gradient analysis and characteristic wave analysis are key issues of the computational theory for solving diffusion wave equations and shallow water equations respectively. The main point to couple these two approaches is to successfully address their connection on the coupling boundary. As discussed above, in existing research only one governing equation is solved throughout the computational domain, but hardly considering the interaction between two kinds of governing equations and resized the area of different computational domain. A reasonable and implementable approach coupling the solution

procedure of diffusion wave equations and shallow water equations is the precondition for establishing DBCM. In this study, the slope gradient analysis is performed to determine the current calculated velocity together with current depth, and the characteristic wave analysis is set at coupling boundary as long as velocity and depth have been provided, no matter it is calculated from hydrologic or hydrodynamic model. Then, flow information exchange at coupling boundary is determined according to the characteristic speed which reflect the propagation of flow state in time and space. This method integrates

hydrologic model and hydrodynamic model into a comprehensive system by means of joining the two core steps of slope gradient analysis and characteristic wave analysis together.

In the proposed DBCM, the coupling boundary position will not keep fixed in advance throughout the calculation process. The location where the slope runoff enters the inundation region varies dynamically, and the flood level can also submerge the original inflow points and regenerate new boundaries. Such alternation is close to natural flow processes. The characteristic

wave theory is used to determine the mass and momentum exchange through the coupling boundary. Compared to the "cascade" operation in UCM, the present DBCM can select a hydrologic or hydrodynamic model simultaneously. When a non-inundation region is larger, the water flow movement is mainly obtained by utilizing the hydrologic model. Whereas, when the inundation region is large, the computational domain is given priority to hydrodynamic model.


## 3 Model validation

The numerical model results from DBCM are compared with the analytical solutions, experimental data, and results obtained from existing numerical models. Considering the complexity of the numerical model schemes used in the hydrologic and hydrodynamic models, the hydrodynamic model performance will be validated in the first stage, and then the DBCM will be verified. As described in 2.2, the numerical schemes of the hydrodynamic model(referred to HM2D in the following section) used in this study have second order accuracy in both time and space.

### 3.1 Oblique hydraulic jump

   The oblique hydraulic jump example is a special flow pattern, with an analytical solution being available in open channel flows, which is often used to verify the capability of the numerical scheme in simulating shock wave formation. When a supercritical flow is deflected by a converging wall at an angel $\theta$, the resulting shockwave forms an oblique hydraulic jump at an angle $\beta$, as depicted in Fig. 6. Both the angles of water surface lines behind the shock wave front can be obtained by analytical solution.

In this study, the upstream water depth and velocity are set as 1m and 8.57m s$^{-1}$ respectively, and $\theta = 8.95°$. The width and length of channel are 30m and 40m respectively. In these conditions, the exact analytical solutions are downstream water depth $D_A = 1.49984$m, downstream velocity $V_A = 7.95308$ m s$^{-1}$, and angle    $\beta = 30°$ (Rogers et al., 2001) when flow reaches a steady state.

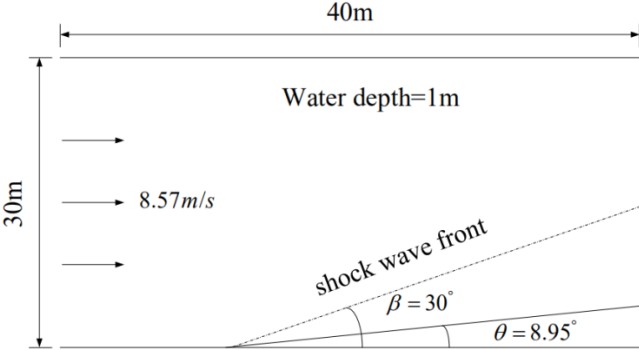

**Figure 6. Oblique hydraulic jump: definition sketch**

   The spatial step size is set as $\Delta x = \Delta y = 0.33$m. The time step is set to dynamic adjustment and total calculating time is 90s. Figure 7 shows a 3-D view water depth results predicted by our model. The oblique jump is sharply captured, and has an angle $\beta \approx 32°$. The average water depth downstream behind the shock front is 1.532m, and the average velocity is 7.86m s$^{-1}$. The numerical solution is close to the analytical solution, as shown in Table 1. The results of references are also shown below. The

output of HM2D and the references, either the water depth or velocity, show good agreements(see Table 1).




**Table 1. Comparison between analytical solution and calculation result for oblique jump case**

|  | Angle $\beta$ | Water Depth(m) | Velocity(m s$^{-1}$) | Depth Error(%) | Velocity Error(%) |
|---|---|---|---|---|---|
| Analytical solution | 30° | 1.49984 | 7.95308 | - | - |
| HM2D results | 32° | 1.532 | 7.86 | 2.1 | 1.2 |
| Reference results | 30° | 1.53 | 7.9 | 2.0 | 0.6 |

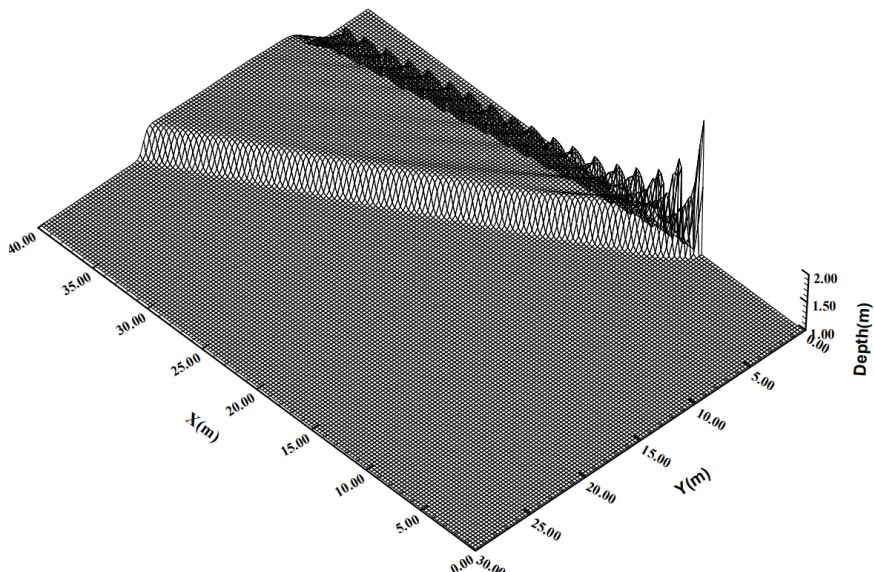

**Figure 7. Steady state of water depth of oblique hydraulic jump**

**3.2 Dam-break over a dry of flood plain**

Dam-break is a classic benchmark problem, which is often used to verify the capability of a numerical scheme in dealing with dry-wet boundary, and the physical experimental model is easy to conduct. Thus, it is convenient to collect measured data for comparison with numerical results. An experiment performed by (Fraccarollo and Toro, 1995) was used to validate the DBCM

developed in this study. The entire model domain is 3m×2m, which is separated into two areas by a dam at X=1m. Initially, the still water with a depth of 0.64m in the reservoir is surrounded by solid walls, while the downstream area is initially dry. The boundaries of the downstream floodplain were all open flow. A 0.4m wide section in the middle of the dam was breached instantaneously. The numerical model spatial step is $\Delta x = \Delta y = 0.04$m, and roughness coefficient is $n = 0.01$.

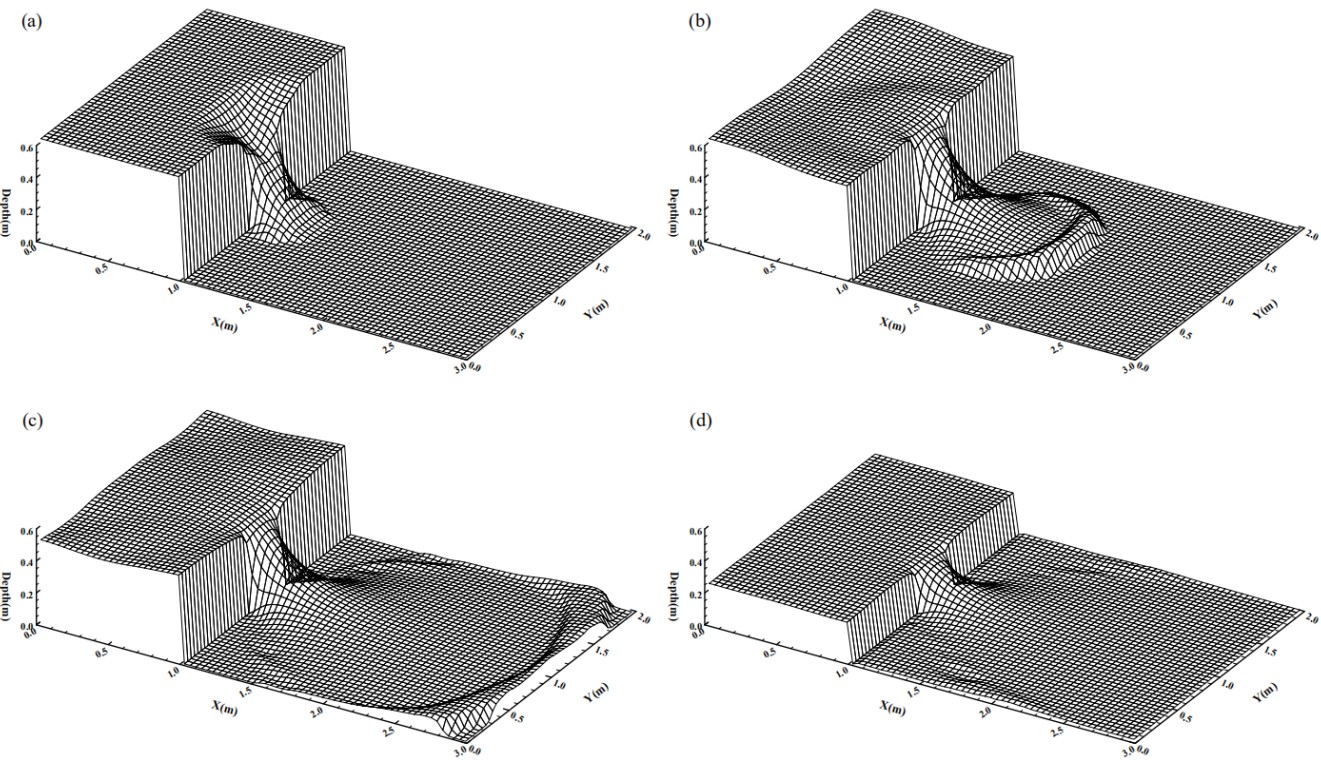

**Figure 8. Snapshot of the water elevation for dam-break simulation: a. t=0.1s; b. t=0.5s; c. t=1.1s; d. t=5.0s**

Figure 8 shows the water surface elevation at different times. It can be clearly seen that as the bore wave propagate toward downstream initially. A depression wave travels upstream, which is reflected back by the walls surrounding the reservoir, causing the water surface elevation in the reservoir to oscillate. In Fig.9, a comparison between the measured and computed water level data was made, which shows a good agreement. The results are encouraging and the overall trend is well captured.



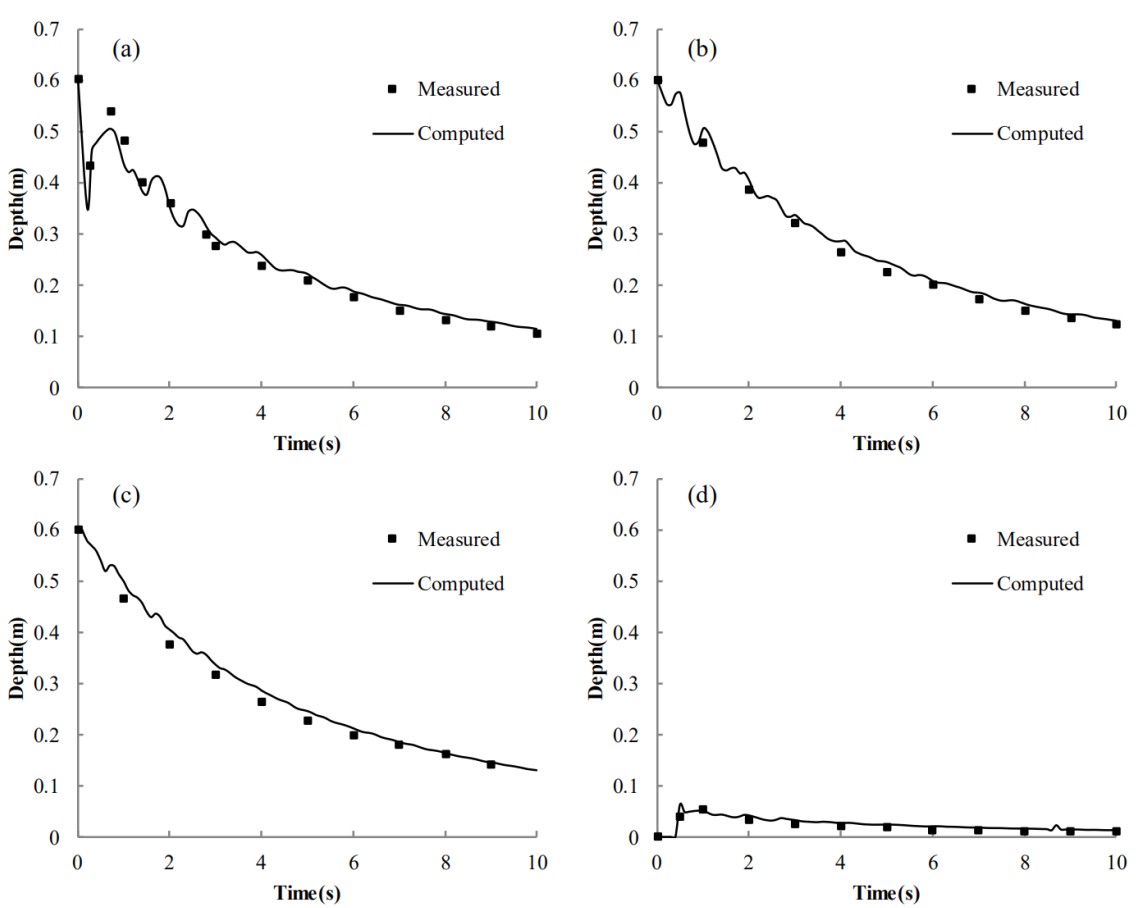


**Figure 9. Comparison of water depth variation at four positions: a. x=1m. y=1m; b. x=0.18m, y=1m; c. x=0.48m, y=0.4m; d. x=1.802m, y=1.45m**

### 3.3 Two-Dimensional surface flow over a tilted V-shaped catchment

A two-dimensional surface flow over a tilted V-shaped catchment is simulated (Di Baldassarre et al., 1996; Panday and

Huyakorn, 2004), we aim to verify whether the computational domains of the hydrologic and hydrodynamic models can

dynamically switch and compare the difference between the DBCM and UCM. As shown in Fig.10, the topography of the

example is depicted.




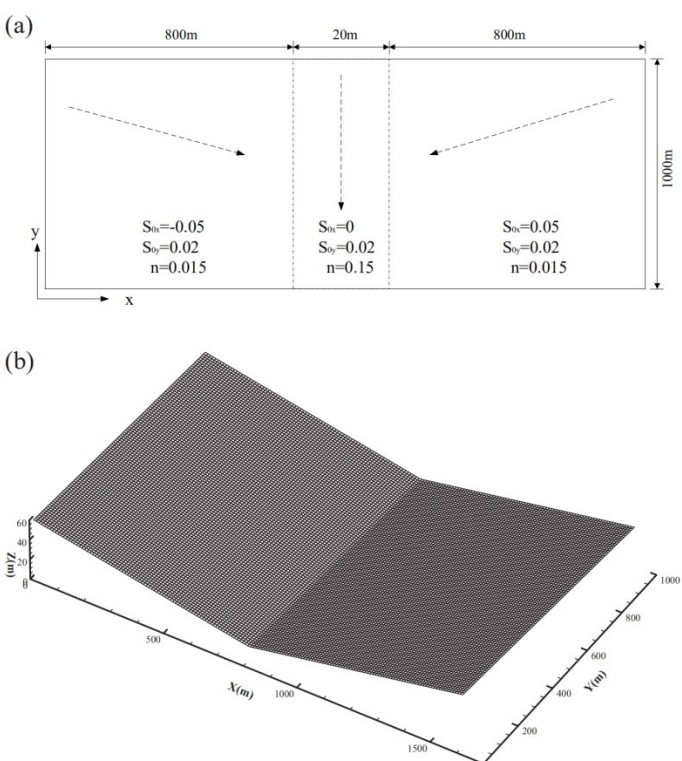

**Figure 10.  A tilted V-shaped catchment: a. dimension of the catchment; b. 3D view of the mesh grid**

The computational domain is a symmetrically V-shaped, with a pair of symmetrical hillslopes forming a channel at central region. The bed lopes are ±0.05 spanwise, and 0.02 streamwise parallel to the channel. The manning coefficient on the hillslope is 0.015 while it is 0.15 in the channel region. Although different from natural conditions, these Manning's values are used to facilitate comparison with other available solutions. The model simulation time is 180min and the rainfall intensity of 10.8 mm h$^{-1}$ over the duration of 90 min is used for the entire domain. Considering the hydrodynamic model providing more details

to describe the overland flow than the hydrologic model, the HM2D and DBCM under the same rainfall conditions were adopted. The water depth threshold distinguishing the hydrological model and the hydrodynamic model is set to 0.005m. In the DBCM the computational region with the water depth being less than the threshold is calculated using the hydrological model, and the region with water depth greater than the threshold is applicable to the hydrodynamic model. The results are compared with four different numerical models developed by (Di Baldassarre et al., 1996; Panday and Huyakorn, 2004), US

Environmental Protection Agency (HSPF) (Johanson and Davis, 1980), and US Army Corps Engineers(HEC-1) (Feldman, 1990). Among these models, HEC-1 cannot handle critical depth(CD) boundary, then the zero depth gradient(ZDG) boundary was specified as the channel outlet boundary. In terms of the other models, including HM2D and DBCM, adopted the critical depth(CD) boundary.



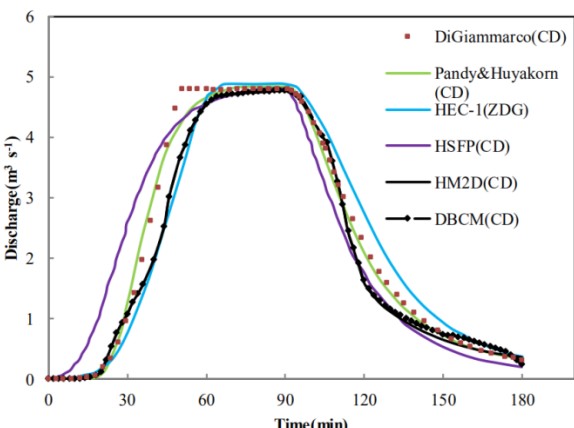

**Figure 11. Comparison of hydrographs simulated by the presented HM2D, DBCM with four different models**

As shown in Fig.11, the outlet hydrographs obtained by the HM2D and DBCM are compared with the other four models. These outlet hydrographs show good agreement for the peak discharge. The start periods of discharge rising and receding limbs simulated by the HM2D and DBCM are consistent with those predicted by others. However, discrepancies gradually grow, so that both the HM2D and DBCM under-predict the discharge. Despite this disparity, the overall trend of the hydrographs indicates that the accuracy of the proposed models are satisfactory.

Comparing the hydrographs between the HM2D and DBCM, it can be seen that their rising limb and peak discharges are in very good agreement. Consequently, both models adopted the hydrodynamic model to simulate the overland flow. The difference between the HM2D and DBCM gradually emerges at the receding limb due to the switching of applied models. The HM2D simulates water movement using hydrodynamic model(shallow water equations) throughout the computation process, while the DBCM switches from the hydrodynamic model to the hydrologic model(diffusion wave equations) when the upstream water depth falls below threshold. Since no time partial derivative terms in the hydrologic model, the velocity at the present is a function of the current water level gradient, and is not equal to the velocity at the previous moment plus the flux term. For this reason, when the DBCM switches from the hydrodynamic model to the hydrologic model, the velocity calculation approach changes accordingly, and the discharge difference between the HM2D and DBCM emerges. Therefore, the outlet flow is slightly larger, but later slightly smaller, in the DBCM, assuring the overall is mass conserved.




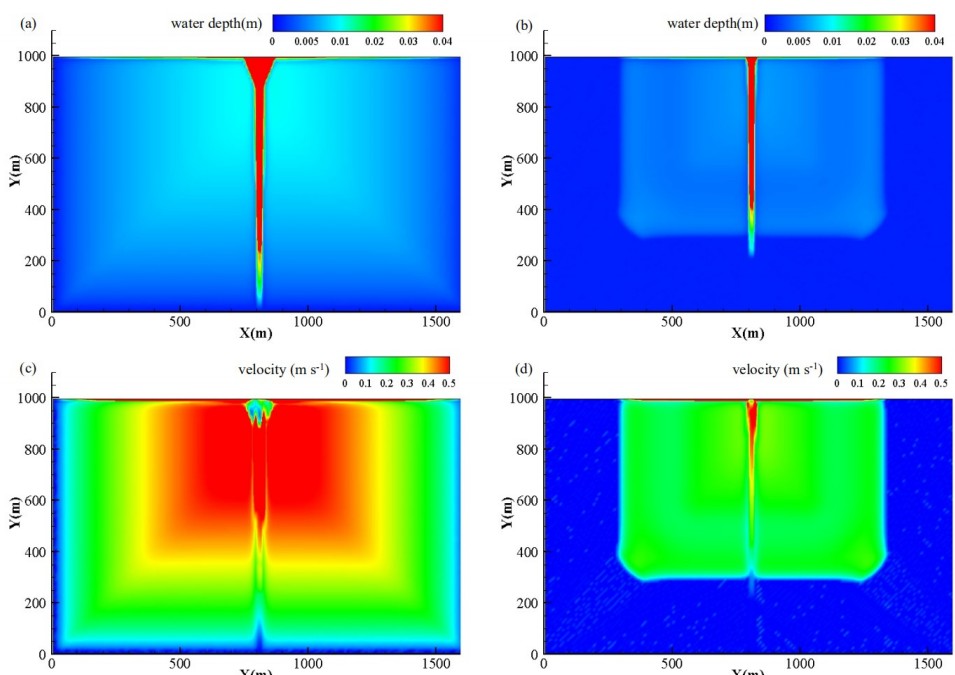

**Figure 12. Water depth and velocity distribution at 90min(a and c) and 120min(b and d)**

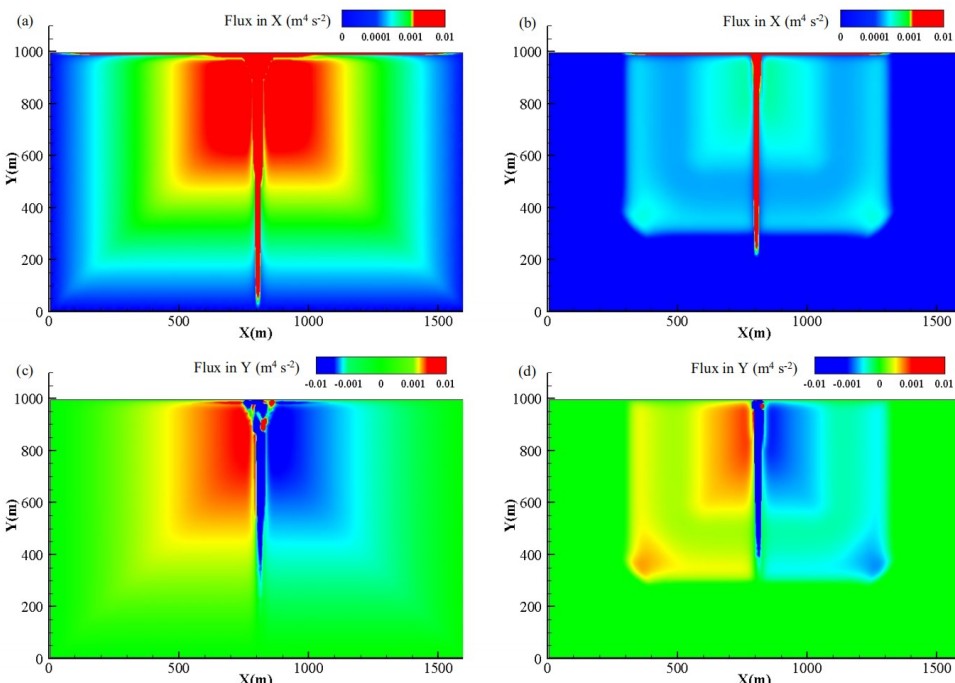

**Figure 13. Flux distributions of in X and Y direction at 90min(a and c) and 120min(b and d)**


The spatial variability of the flow at 90 min and 120min are shown in Fig.12. The hydrological model and the hydrodynamic model are solved simultaneously in the DBCM. The main difference between the governing equations of the hydrologic model and hydrodynamic model is that the flux term is not calculated in the former, meanwhile the latter needs to calculate the convection term. The non-inundation region and inundation region can be determined by whether the flux term is generated during the calculation process. At t=90min, the rain stops, the water depth is the highest and, the flow state is calculated by the

hydrodynamic model over the whole domain. At t=120min, the water continues to flow to the outlet and the water near the upstream region decreases, but a small amount of water still exists. No flux is calculated while velocity computation continues. Obviously, a sharp division line separating the domain arises at this moment.

        The results from the UCM is compared with those by the HM2D and DBCM. Both the HM2D and DBCM use rainfall as the boundary condition and simulate the flow movement in the whole domain. The UCM employs the outlet hydrographs

obtained by the HM2D as the upstream inlet flow and lateral flow boundary condition respectively to calculate the flood movement in the channel. At first, the hydrographs of UCM adopting the inlet flow boundary condition are compared with that of HM2D and DBCM at the inlet and at outlet, as shown in Fig.14. Note that the upstream discharge is calculated by the HM2D and DBCM are both very small, which are several orders of magnitude smaller than that by the UCM. Both the results by the HM2D and DBCM are close to the actual situation. After the rain falls in the upper reaches of the basin, water flows to

the lower reaches very quickly. The overland flow in the downstream region includes the local rainfall-runoff and surface runoff from upstream. The closer the region is to the downstream, the greater the surface flow. However, the UCM takes the outlet hydrographs as the upstream inlet boundary condition to computer channel flow, which amplifies the upstream flow significantly. The outlet flow also appears differently, not only the time lag of flood peak, but also the reduction of peak discharge and the flood waveform changed. It can be concluded that the UCM overstates the disaster in the upper reaches of

the basin, and the overflow in the downstream reach is small, thus the arrival time of the flood peak will be inaccurate.

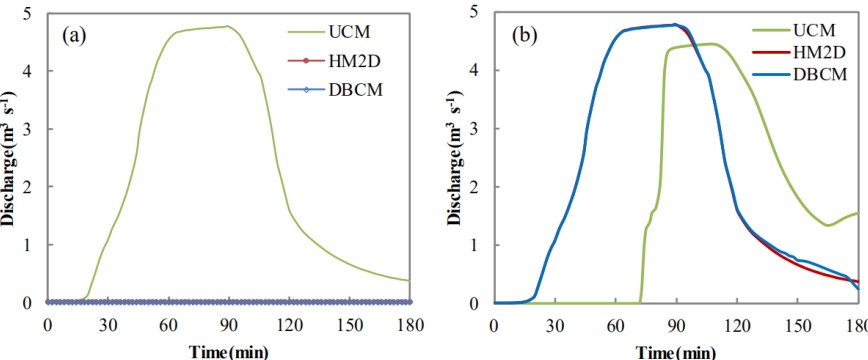

**Figure 14. A comparison of hydrographs at upstream(a) and outlet(b) for upstream inlet boundary condition**

        Figure 15 shows the UCM results when adopting a lateral flow boundary condition, in which the flow is distributed evenly throughout the channel. The upstream flow is reduced, but a significant gap between the UCM, HM2D and DBCM still exists.





Even though the peak flow is almost equal to the results from the HM2D and DBCM, the outlet discharge obtained by the
       UCM with lateral flow boundary is biased, no matter the arrival time of flood peak or flood waveform..

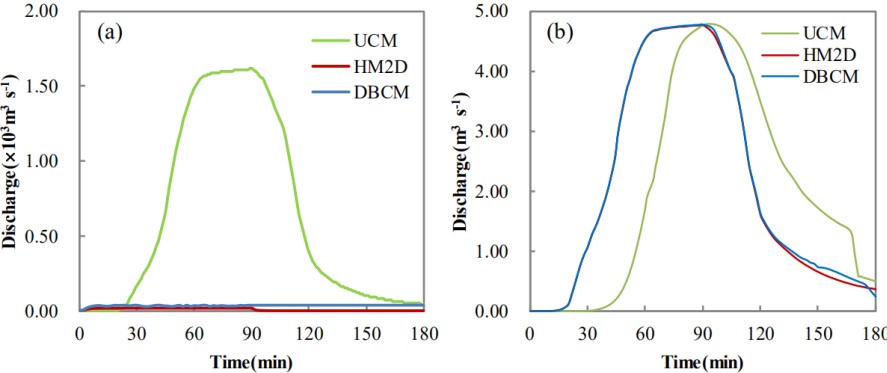

**Figure 15. A comparison of hydrographs at upstream(a) and outlet(b) for lateral flow condition**

       Finally, the need for model transformation is discussed. Flood propagation is a phenomenon of high speed movement with
drastic change of water depth and velocity. The hydrologic model(diffusion wave equation, omitting convection term) is
       insufficient to describe this movement. Fig.16 depicts the rapid change of water depth profile near the outlet in a short time,
       while the water depth on both sides of hillslopes hardly changed. This leads to strong convective flow near the channel, and
       the application of hydrologic model inevitably leads to errors.

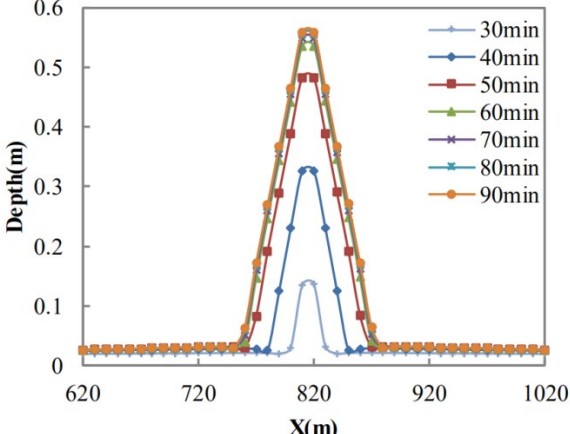

**Figure 16. A comparison of water depth profile in the channel outlet**

**4 DBCM implemented to a natural watershed**

The proposed DBCM was used to undertake a flood risk assessment to a natural watershed, i.e., Helin Basin of the Longxi
River in Chongqing City. The Longxi river basin is located in the eastern region of Chongqing, which is a first-class tributary
of the Yangtze River. The main channel is about 221km long and the basin area is about 3280km². The overall terrain gradually
goes down from northeast to southwest, consistent with the trend of the main channel. Most of the central and southwest areas

are plain, and the east and west areas are mountainous, a typical topography of a trough sandwiched by two mountains. The average annual rainfall in the basin is 1192.4mm, which is prone to heavy rain in summer, and the flood spreads rapidly to the central plain as a consequence of the topographic feature. The selected catchment, Helin basin, located in the northeast of the Longxi river basin, was chosen as a case study for investigating the surface flow phenomena using the DBCM and UCM. The

location of the Helin basin is shown in Fig.17.

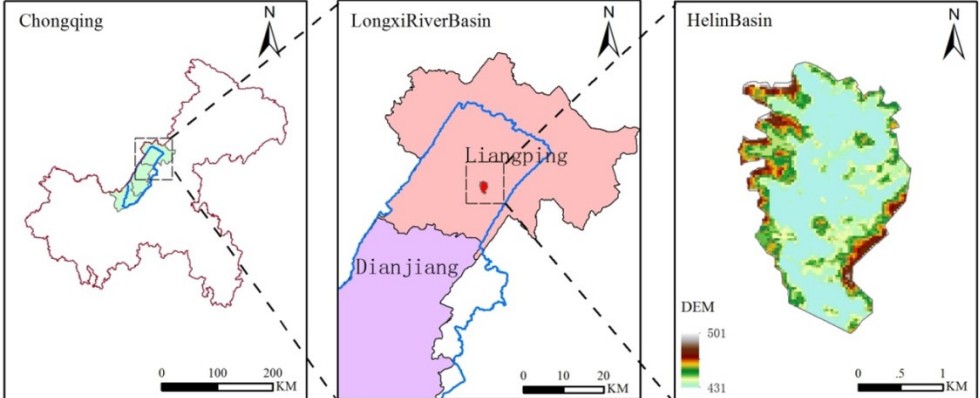

**Figure 17. Location of Helin town. Chongqing province(left), Longxi river basin (middle), Helin basin(right)**

The river section in Helin basin is a typical mountainous river. The upper part of the river has a steep slope, while the middle and lower reaches are relatively gentle. The river banks is in the raw  without flood protection works. The terrain along the

river is open plain, and the farmland is widely distributed, resulting in the poor ability to resist flood disasters. Once the flood level rises, the river bank collapses and floods overflow due to the erosion of the bank slope with high flow rate, resulting in serious collapse of the existing natural river bank and serious damage to the coastal crops. The flood is caused by the rainstorm, and the flood season is consistent with  the rainstorm season which lasts from April to September. Heavy rainstorms and flood often occur during this period.

The model data includes DEM, LULC(land use and land cover) and soil type as shown in Fig.18. DEM data were obtained from the GDEMV2 database with spatial resolution of 30m. The DEM was resampled according to some channel section field survey data to get finer resolution. There are four main kinds of land use in Helin basin: urban, forest, farmland and water. Besides, several soil types with little proportion have been consolidated into the categories with a large ratio. Soil properties determine the infiltration rate, which further affect the surface runoff. The parameters, such as roughness and soil moisture

content, are extracted from the public data. The LULC data are processed by remote sensing interpretation tools using the satellite image. The soil data are processed by SWAT, Soil-Plant-Air-Water (SPAW) model using the Harmonized World Soil Database (HWSD). Then the collected parameters are  conformed according to the reference opinions for hydraulic engineering construction provided by local water conservancy department.




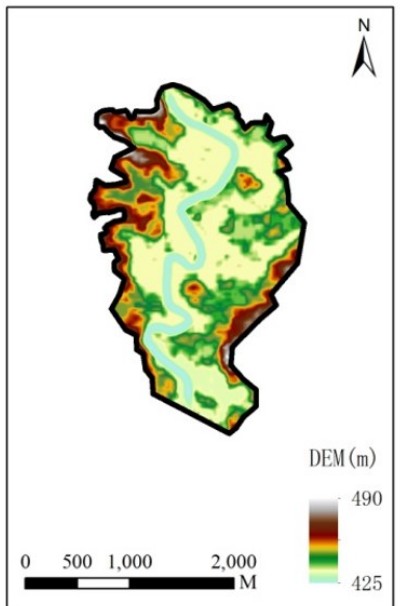
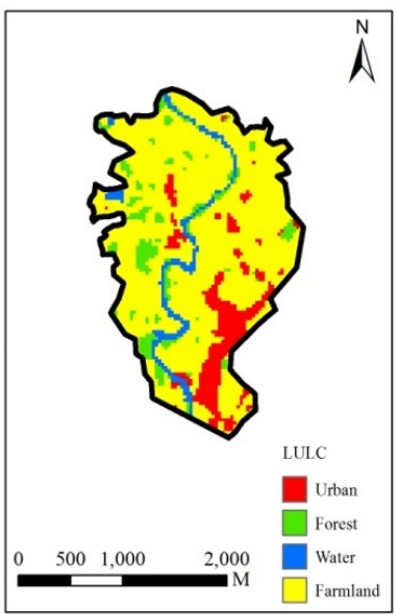
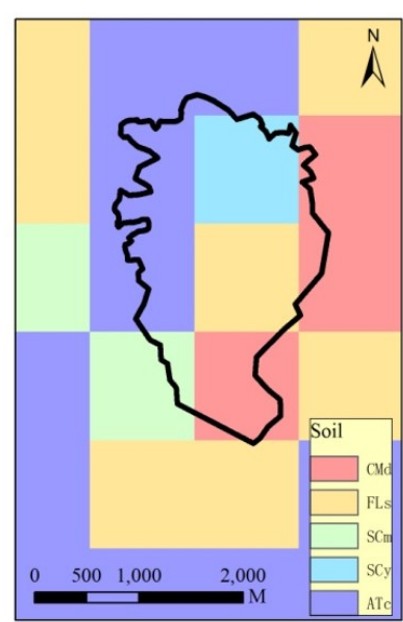

**Figure 18. DEM, LULC and Soil of Helin basin**

The DBCM was applied to model the rainfall runoff process in the Helin basin and the result was compared with a UCM composed of SWAT(as hydrologic model) and HEC-RAS(as hydrodynamic model). The SWAT was calibrated by the design storm and hydrographs (flood return period of 1%) in the Helin basin outlet. The selected coefficients of rainfall and flood are listed in Table 2 and Table 3. The calibration results are shown in Fig.19. It can be seen clearly that the flood process calculated by the SWAT model is very similar to the design flood, with similar flood peak flow and fluctuation process. In the early stage, there is no surface runoff although the rainfall lasts for a short time, as a result of the soil infiltration. Later, the discharge curve climbs rapidly, consistent with the rainfall intensity, on account of the saturated soil. The simulation result reflects the storm runoff production process affected by the combined action of rainfall and infiltration. The hydrographs show a good agreement with the design flood, demonstrating that SWAT has been calibrated and its output are reliable for hydrodynamic model.

After calibration, the hydrographs computed by SWAT in different locations were extracted and applied as inflow boundary conditions for different models. Three simulation scenarios are designed, as shown in Table 2. The first two cases are used to verify the proposed model to real catchment, while the third one is employed to compare the DBCM with the UCM.

**Table 2. Simulation scenarios**

| Case | Model | Boundary condition | Descriptions |
|---|---|---|---|
| A | HM2D | Helin out flow as inflow BC | HM2D is a kind of UCM proposed in the present study |
| B | HEC-RAS | Same as Case A | HEC-RAS is also a kind of UCM |
| C | DBCM | Helin inlet flow | The coupling model proposed in this study |




Table 3. Rainfall parameters (Cs: coefficient of skewness Cv: coefficient of variation P: flood recurrence period)

| Duration (h) | Mean value (mm) | Cv | Cs/Cv | P(%) | | | |
|---|---|---|---|---|---|---|---|
| | | | | **1** | 2 | 5 | 10 |
| 6 | 81.7 | 0.5 | 3.5 | **224** | 197 | 162 | 137 |
| 24 | 112 | 0.48 | 3.5 | **297** | 263 | 218 | 183 |

Table 4. Peak discharge at Helin outlet for different flood frequency

| P(%) | **1** | 2 | 5 | 10 | 20 | 50 |
|---|---|---|---|---|---|---|
| Discharge (m³ s⁻¹) | **2280** | 1920 | 1470 | 1150 | 831 | 433 |

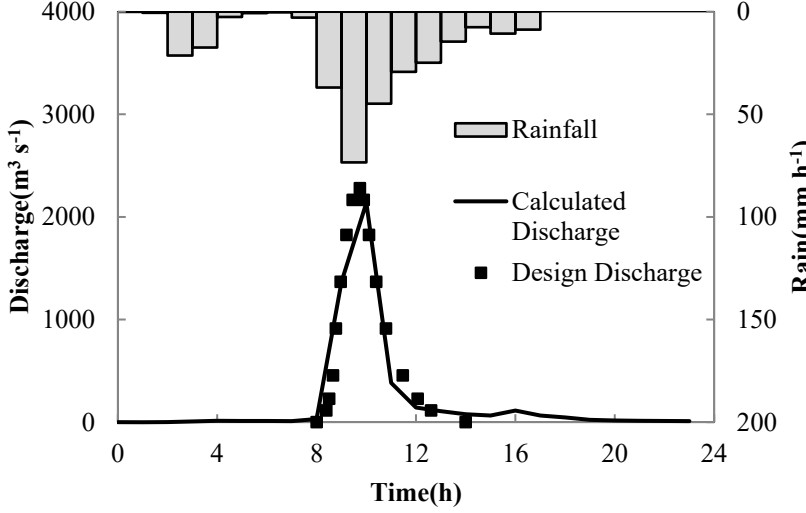

Figure 19. Design rain and flood with calculated flow in Helin basin outlet(1%)

The maximum water depth distribution and hydrographs at three positions ($p_1$, $p_2$ and $p_3$) are used to validate UCM and the DBCM model. The calculation results of case A and case B are shown in Fig.20 and Fig.21. In the scenario for the flood of once in 100 year (P=1%), the low areas in the Helin basin are almost completely submerged, and the maximum submerged depth in most areas is between 2 and 5 meters, while the scattered highlands are relatively safe, which can be reflected in the predictions using the HM2D and HEC-RAS. In Fig.21, the water depth profiles of HM2D and HEC-RAS at three locations also show good agreement. Note that the profile descends to a crest in a later period, then rises to a slightly higher level, which is attributed to the sampling point location and terrain topographic influence. The three points all locates in the riverbed lower than surrounding land. Recalling the model predicted inflow hydrograph in Fig.19, a tiny discharge crest appears in latter part of the simulation time and then falls to zero. In Fig.21a, this small trough still exists in the upper reach and keeps the same pattern as the inflow, but vanishes in the middle and downstream reaches due to surface resistance in Fig.21b and Fig.21c. In




the early stage of simulation, a large amount of water flows out of the channel and floods the plain area. Later on, the inflow gradually falls resulting in that no water supplement from upstream enters reaches, hence the crest arises. Soon, the water inundating plains flows back into the river gradually because of elevation difference, then water depth rises and keep steady. Either the global or local distribution of water depth has demonstrated that the proposed HM2D gained satisfactory results 475 compared with HEC-RAS, and HM2D(as part of DBCM) can be applied in practical engineering projects with complex terrain.

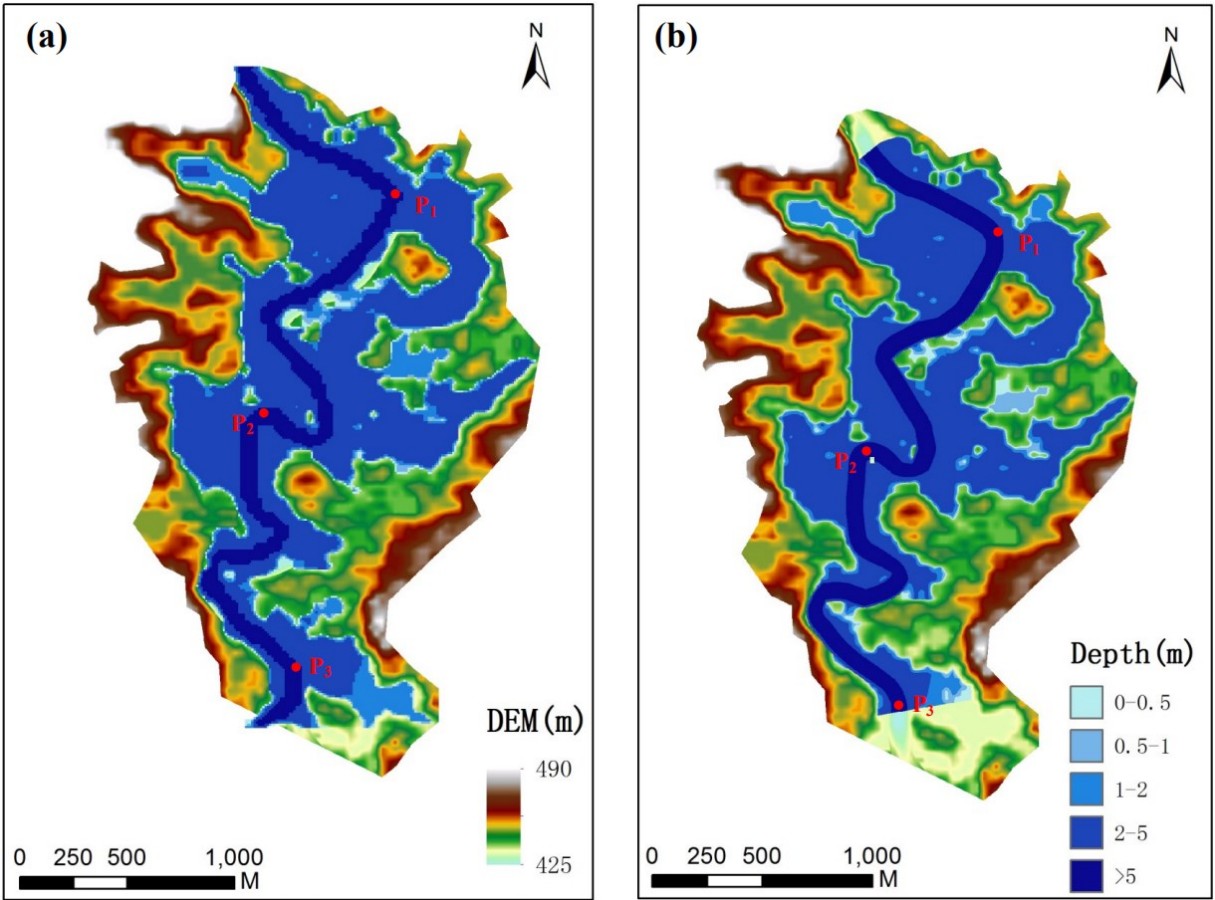

**Figure 20. Comparison of maximum depth, (a)case A, (b)case B. Red points denote 3 locations to compare water depth**





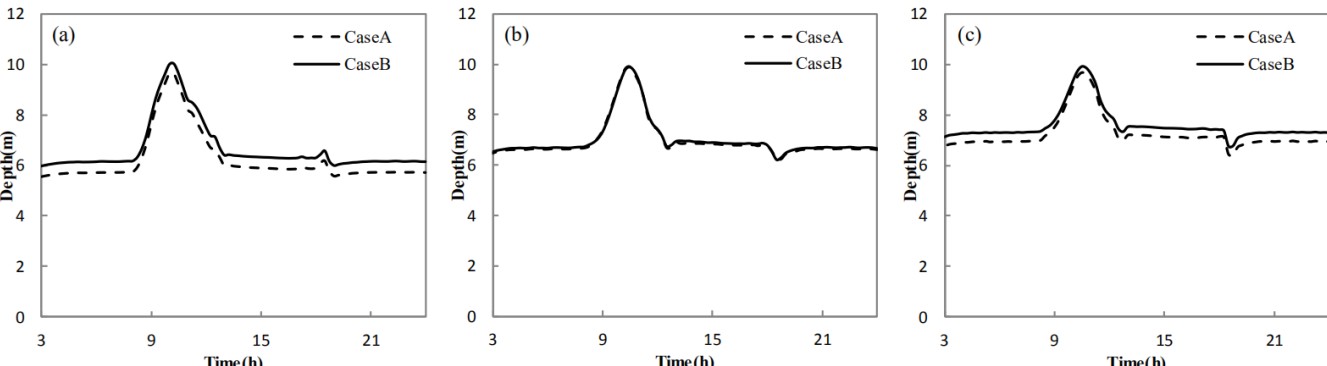

**Figure 21. Comparison of water depth at 3 positions, (a)P$_1$, (b)P$_2$, (c)P$_3$**

The results from Case A and Case C are compared to investigate the capability of the DBCM. Fig.22 depicted the maximum water depth distribution in Case A and Case C. The inundation region has expanded significantly in Case C. Not only the lowland areas, but the hillsides have been inundated. Even though Case A adopted the simulated outlet hydrographs larger than the one in inlet used in Case C, the runoff failed to inundate the hillside because of topographic obstruction. The red cycle in Fig.22 indicates the urban area. Water depth was extracted for each cases. Due to the lack of measured data, field survey

and historical records have to be used as reference data to verify the model outputs. The current river bed conditions are as follows: both sides of the river are flat, with a lot of farmland and some villages distributed along the main river. No embankments or bank protection works have been built along the river course. All of these problems lead to low flood control capacity of the reach. As local residents recall, in 2017, the flood covered the middle of the trees along the bank, equivalent to at least 3m water depth. Reviewing the simulation results(Fig.20), the maximum water depth is generally between 2m and 5m

along the banks. When it comes to urban area, according to the historic record, the rainstorm in 12 August 1998 caused a flood that local streets and airports were inundated with water depth of 1.0m and 1.4m respectively. All villages and towns in the Helin town catchment were submerged with water depth exceeding 0.5m on average. In Case C, this phenomenon was simulated that the maximum water depth in urban areas is more than 0.6m, in accord with the historic data. But no water emerged in Case A, although the Helin outlet flow is utilized as inflow discharge, greater than that of Case C, as shown in

Fig.23. Referring to local topography, the main cause of this problem is that urban locates in a higher position, between riverbed and hillside, hence the upwelling movement of water in rivers is easily blocked by terrain. Nevertheless, urban area will be submerged by flow from the uphill slope, even though the river flow has been obstructed. Obviously, the computed results of Case C by the DBCM are approximate to the practical situation.




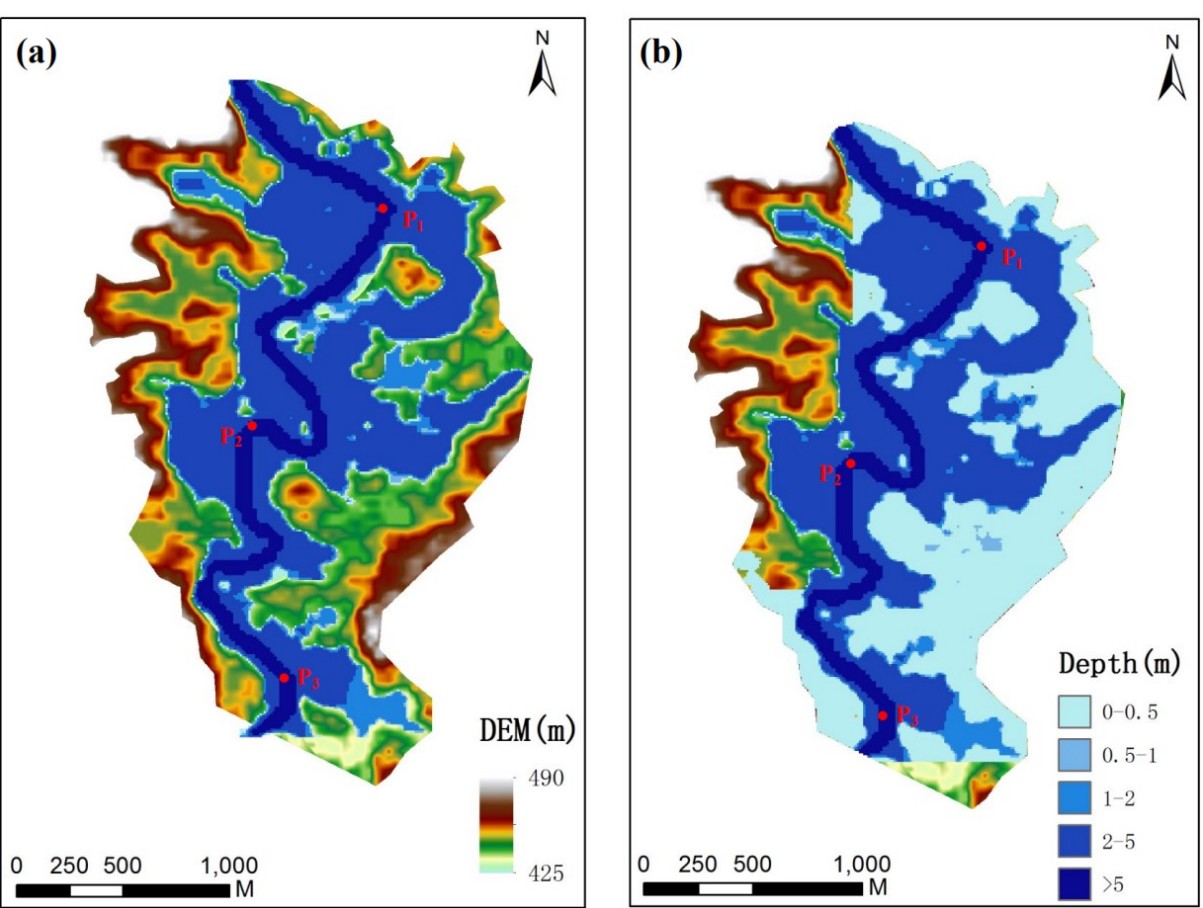

**Figure 22. Comparison of maximum depth, (a)case A, (b)case C. Red points denote 3 locations to compare water depth**

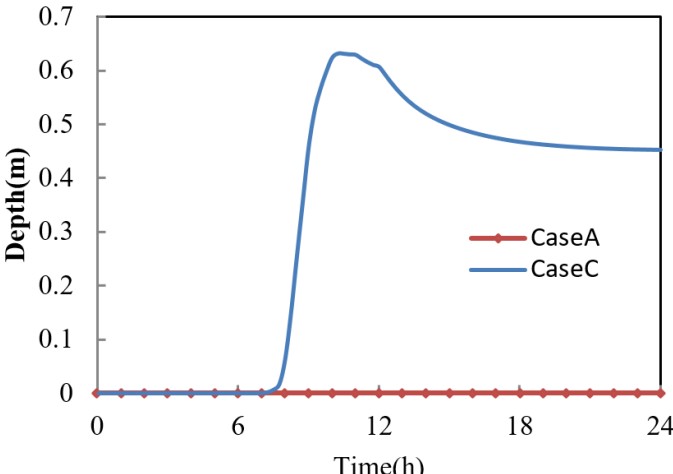

**Figure 23. Comparison of water depth variation**





The capability of DBCM for switching models dynamically during computational process is tested, and the parameters of water depth, velocity and flux term are selected to conduct this assessment. The water depth threshold(0.01m) is used to
distinguish the two models. Velocity is calculated in all of the inundation region, while the flux term is computed only in the region where hydrodynamic model is used, not the hydrologic model region. As shown in Fig.24, the rainfall stopped after 17h since the simulation started, and the surface flow on the slope gradually decreases. Flux calculation no longer exists in most part of the slope regions. However, a small amount of water is still left on the sloping area, and the flow in the confluence area is calculated by the hydrologic model(diffusion wave equations). Thus, even though flux calculation has stopped, flow
velocity still exists in most of the slope. The low-lying area at the northeast corner, due to the obstruction of the terrain, cannot be flooded in Case A. However, through the DBCM, the confluence of the surrounding slope accumulated to the local area and form a small range of flooded area.



**Figure 24. Simulation results at t=17h, (a) and (b) are water depth(m) for case A and case C,(c) and (d) are velocity(m s$^{-1}$**

**) for case A and case C, (e) and (f) are flux term(m$^4$ s$^{-2}$) for case A and case C**



# 5 Conclusions

A dynamic bidirectional coupling model (DBCM) for flood prediction and analysis was developed. The mathematical formulations and the solution schemes of the DBCM are realized. In DBCM, the runoff production is depicted by the hydrologic model through the rainfall-runoff process, while the hydrodynamic model study emphasizes on the flood
propagation processes. The characteristic wave theory is applied to compute the coupling boundary between the hydrologic and hydrodynamic computational domains.

In using the proposed DBCM, a dynamic change of the boundary position is realized for determining the non-inundation and inundation regions, which enables the mass and momentum exchange and interaction between the two regions. The hydrologic and hydrodynamic model are carried out simultaneously. The DBCM is more in line with the natural physical
process of flood formation and propagation, which has the potential to significantly improve the accuracy of flood prediction.

The main advantage of the proposed DBCM are:(1) Based on the characteristic speed theory to predict flow propagation. The DBCM realizes the discharge calculation at coupling boundary by coupling slope gradient analysis and characteristic wave analysis, which are the foundations for solving hydrologic model and hydrodynamic model respectively. (2) The discharge at coupling boundary is used to update conservation variables of hydrologic model and hydrodynamic model with explicit
physical significance based on the consideration of the flow state on both sides of the coupling boundary.(3) Contrast with the UCM and BCM, where the computational domains for hydrologic and hydrodynamic models are independent of each other and remain fixed, the DBCM can resize the computational domains of inundation and non-inundation regions according to the flow state throughout the calculation process, which is more aligned with natural rainfall and flood propagation conditions.

Three test cases show that the DBCM is capable of accurately simulating the hydrologic and hydrodynamic response to
rainfall events in various catchments. The DBCM gains good agreement with the analytical solution, and realizes the switching between the hydrologic and hydrodynamic models in simulating overland flow, hardly achieved by former methods, with single model working. The DBCM also succeeds in predicting the inundation regions in natural storm flood events with more precise results when compared with the UCM.

## Data availability

Model simulation and calibration data are available upon request from the corresponding author. Digital elevation model data are provided by the Geospatial Data Cloud at http://www.gscloud.cn. The data sets of Soil Properties and Land cover are provided by Cold and Arid Regions Sciences Data Center at Lanzhou (http://westdc.westgis.ac.cn).



## Author contributions

YWY designed the methodology and carried out the investigation with cooperation from all co-authors. QZ provided the
original model input data. The study was supervised by CBJ. YWY prepared the first draft of the manuscript, which was then
revised and improved by the co-authors.

## Competing interests

The authors declare that they have no conflict of interest.

## Acknowledgements

This study was supported by the National Science Foundation of China (No. 51679121) and National Key Research and
Development Program of China(NO. 2016YFC0502204).

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
