# Peer review of "A Dynamic Bidirectional Coupled Hydrologic-Hydrodynamic Model for Flood Prediction"

_Natural Hazards and Earth System Sciences, 2019_

## Referee Comment (RC1) · Anonymous Referee #1 · 28 Jan 2020

The manuscript is interesting coupling between two different models to improve flood simulation.

My first comment is directed to the title. Flood prediction is stated as the main aim. This should be rephrased to flood simulation to avoid confusion. Prediction is often associated with forecast, which is not the aim of this manuscript.

My second comment is related with the branding of "2D diffusion wave" with "hydrological model". It seems that the authors have developed a 2D diffusive wave model (line 139). If that is the case, this cannot be categorised as hydrological model. The title should be rephrased to "coupled diffusive-full dynamic". Unless the authors can justify the branding of hydrological model, this must be changed.

[Figure]

Some properties of Hydrological models are: a) only propagate information downstream; b) are inherently one-dimensional when simulating flood routing (channels or links), and c) able to simulate other discharge components such as interflow and baseflow besides direct discharge.

Hence it seems that this not fits to the description of the authors. This requires some rethinking and restructuring of the manuscript, but is required in order to avoid misinterpretation of the good work developed.

My third comment is related to the time steps. how are the time steps being calculated, and how are the two model synchronized?

My forth comment is related with the display of the results. The manuscript is about coupling two different models; however, it is not clear in the plots where the boundary is. Please add to all plots the location of the boundary between the two.

minor comments: line 60, "evolved" instead of "involved"

line 60, "overestimate the flood risk in some extent" is too vague, please rephrase

line 70 to 73, is confusing. e.g. the same sentence starts with "the next step", and ends with "the next time step", is the former not time? What is the meaning of "present flow state"? is that same as current, as stated previously? if yes, always use same wording for the same meaning.

line 80, why is it a "significant problem" please explain.

line 85, remove "of"

line 86, should read "considers"

line 91, "apt to", replace with "adequate". to perform what? (not clear). "doesn't" replace with "does not"

line 82 should read "further studies are necessary"

line 109, what is "slope runoff" - consider removing the word "slope" and use simply runoff throughout the text

line 173, is telemac being used, or it was "re-written". please clarify this.

line 193, is the hydrological model 1D or 2D, if it is the latter, it can also produce inundation extents. please add some plots which show the boundary of the "hydrological" and hydraulic model being changed. Overlay these with the flood inundation extent.

line 212, "is moved to point A", so where was it before?

lines 215 to end of paragraph is confusing.

Figures 2 and 4, what is the colour code?

Figure 4, why water cannot flow to the left in the middle figure?

Figure 7, why are there spikes on the depth?

Figure 8, is this already with the DBCM? if yes, where is the boundary of the two models in those plots.

---

## Referee Comment (RC2) · Anonymous Referee #2 · 21 Feb 2020

The manuscript presents a novel dynamic coupling approach of hydrologic-hydrodynamic simulation, where representation of flood processes and simulation can be potentially improved, and therefore a substantial research topic. However, unfortunately the performance assessment of the model has not been done properly and extensive enough to justify the strength of the proposed coupling scheme. For instance, the performance is only compared to UCMs, instead of BCM, as that's where the novelty value of DBCM lies upon and should be evaluated. Meanwhile, in the comparison to UCMs (section 4), the hydrological consideration using point source inflow boundary is not an appropriate method to support authors' claim (see below point 3).

I also think that the presentation of the manuscript can be much further improved in terms of language clarity, and figure presentation.

[Figure]

More specific comments can be found below: 1) In Section 4, Scenarios in table 2 and their related text: It is not sure how the result from the hydraulic model in Case B being transformed into the spatial distribution of water depth, e.g. in figure 20.b. Is the HEC-RAS setup, i.e. in case B, considers both 1D & 2D unsteady flow simulation (i.e. in HEC-RAS v.5) OR just 1D + GIS "bathtub method" OR just 2D? Also please provide at least table or information for these setups in the supplementary so that it benefits others who may want to compare in the future.

2) I would also consider improving the presentation of figures 20 & 22, e.g. to remove the information of the elevation in the backgrounds and they may only confuse readers with many colors. Instead, since the authors validated vaguely with the record of water depth reported in the urban areas, the addition of extent, e.g. hollow polygon of these urban areas would be more useful. This could also point out further the claim of the author regarding case A failing to simulate flood in the urban areas (lines 488 -495), or rather parts of the urban areas (higher elevation).

3) Line 495-498, with regards to mentioned reasoning of case A and B failing to simulate the flood depth at the higher elevation further from the river bed, I would add the obvious reasoning is the fact that case A and B contains NO distributed hydrologic modelling, they only consider point sources inflow boundary conditions obtained from SWAT, and therefore none of the spatial rainfall distribution is considered. Such failure has NOTHING to do with the lack of dynamic nor bi-directional coupling method. Therefore, the author's approach for the evaluation/ comparison is not appropriate to prove the strength of the author's DBCM.

4) Since the BCM coupling model (i.e. MIKE SHE + 11) is described timely and only compared conceptually along with author's proposed DBCM and UCM, why is the DBCM is only compared with UCM in terms of performance (instead of all 3)?

5) The performance of bi-directional against uni-directional coupling for flood modelling has already been compared/accessed and known overtime in literatures for its improved water transfer dynamic representation and result when setup properly. Since the author's novelty for the coupling approach is emphasize on the improved representation of changing extend of boundary or in other words spatial-dynamic on top of the BCM, I would say the strong focus of the finding should be on the performance assessment of author's DBCM over BCM instead of the obvious UCM vs DBCM.

6) Line 115, . . .involves the "processes of precipitation"? What does it mean within the context?

7) Line 430-440 and Figure 18, not very clear, please provide a better legend for the soil type, what they mean instead of meaningless abbreviation only and imply for the model, e.g. relating to SCS number or others. Also, please consider providing appropriate reference & source to data inputs like GDEMV2, LULC, and soil type database.

8) Line 443, The selected coefficient of rainfall and floods. . . in table 2 & 3, I think you meant Table 3 & 4 instead. Also 'coefficient of flood' is not appropriate.

9) Line 483, "The red cycle. . .urban area"? Even if it is "circles/ points", I understand they are discharge comparison site (p1, 2 & 3).

Minor: Figure 17 & 18, the DEM information/ figures are repeated, and besides they are also shown differently/ not consistent.

Figure 19, I suggest "Simulated Discharge (SWAT)" instead of "Calculated Discharge".

Figure 20 & 22, considers revising the legends (incl. the scale) into one single space since they are the same rather than repeated in 2 separate figures, currently It may give the impression that only elevation legend applies to figure a while depth only to figure b.

Table 2, "outflow" instead of "out flow".

Line 340, "slopes" instead of "lopes".

---

## Author Comment (AC1) · 10 Apr 2020

**Answers to Anonymous Referee #1 comments**

We would like to thank Referee #1 for the constructive comments, very detail corrections, and recommendations towards improving our manuscript. We are improving our writing quality based on your kind suggestion. These comments are all valuable and very helpful for improving our paper. We appreciate that we have a chance to revise the manuscript as you recommend and to resubmit our manuscript will meet your approval.

In the following, we respond point by point to the comments. The referee comments appear in black and the answers appear in blue.

1/ My first comment is directed to the title. Flood prediction is stated as the main aim. This should be rephrased to flood simulation to avoid confusion. Prediction is often associated with forecast, which is not the aim of this manuscript.

The title has been changed into "A Dynamic Bidirectional Coupled Surface Flow Model for Flood Inundation" combined with consideration of the second comment.

2/ My second comment is related with the branding of "2D diffusion wave" with "hydrological model". It seems that the authors have developed a 2D diffusive wave model (line 139). If that is the case, this cannot be categorised as hydrological model. The title should be rephrased to "coupled diffusive-full dynamic". Unless the authors can justify the branding of hydrological model, this must be changed.

Some properties of Hydrological models are: a) only propagate information downstream; b) are inherently one-dimensional when simulating flood routing (channels or links), and c) able to simulate other discharge components such as interflow and baseflow besides direct discharge.

Hence it seems that this not fits to the description of the authors. This requires some rethinking and restructuring of the manuscript, but is required in order to avoid misinterpretation of the good work developed.

Indeed, the method proposed in the manuscript was mainly on the coupling of a 2D diffusion wave model and a full-dynamic model. Whereas, the model developed includes the hydrological processes of precipitation, infiltration and runoff routing, so we thought it belongs to a hydrologic model which will include other hydrological processes in the future, such as, evapotranspiration, snowmelt, satuated zones, etc.

3/ My third comment is related to the time steps. how are the time steps being calculated, and how are the two model synchronized?

Time steps are determined by CFL condition as following:

$$\Delta t = C_r \min\left(\frac{\Delta x_i}{|u_i| + \sqrt{gh_i}}, \frac{\Delta y_i}{|v_i| + \sqrt{gh_i}}\right)$$

This formula has added to the end of section of 2.2 Hydrodynamic model in the modified version.

4/ My forth comment is related with the display of the results. The manuscript is about coupling two different models; however, it is not clear in the plots where the boundary is.

Please add to all plots the location of the boundary between the two.

The evolvement of coupling boundary has been added to the V-shape catchment case and Helin Basin case in the modified manuscript.

5/ minor comments:

line 60, "evolved" instead of "involved"

Thanks for correction

line 60, "overestimate the flood risk in some extent" is too vague, please rephrase

The text was rephrased in the modified manuscript.

line 70 to 73, is confusing. e.g. the same sentence starts with "the next step", and ends with "the next time step", is the former not time? What is the meaning of "present flow state"? is that same as current, as stated previously? if yes, always use same wording for the same meaning.

The text was rephrased in the modified manuscript.

line 80, why is it a "significant problem" please explain.

The problem indicates the bargain of cost and benefit between simulation efficiency and resources as illustrated the following text.

line 85, remove "of"

Thanks for correction

line 86, should read "considers"

Thanks for correction

line 91, "apt to", replace with "adequate". to perform what? (not clear). "doesn't" replace with "does not"

Thanks for correction

line 82 should read "further studies are necessary"

Thanks for correction

line 109, what is "slope runoff" - consider removing the word "slope" and use simply runoff throughout the text

Thanks for suggestion.

line 173, is telemac being used, or it was "re-written". please clarify this.

TELEMAC is not used since it is developed based on unstructured grid, while the model in this paper is based on Cartesian grid.

Change in manuscript: These equations are solved using the finite volume method similar to TELEMAC (Ata et al., 2013). And the convection flux on grid faces is calculated using the HLL scheme with WAF approach (Toro, 2001).

line 193, is the hydrological model 1D or 2D, if it is the latter, it can also produce inundation extents. please add some plots which show the boundary of the "hydrological" and hydraulic model being changed. Overlay these with the flood inundation extent.

The 2D diffusion wave equations were used for the runoff routing process, so it is 2D.

The time varied coupling boundary evolvement has been added to the modified manuscript.

line 212, "is moved to point A", so where was it before?

This indicates the coupling boundary position at the present time step. For the first case, the coupling boundary position will moved to A in the next step, as shown in the following figure(a).

[Figure]

lines 215 to end of paragraph is confusing.

The text has been rewritten in the modified manuscript.

Figures 2 and 4, what is the colour code?

Figure 2 and Figure 4 will be merged into single one as follows:

[Figure]

Figure 4, why water cannot flow to the left in the middle figure?

The effects of gravity are that the flow at any point will trend to be in the direction of the steepest water surface slope, as shown in the following figure:

[Figure]

For clarification, figure 4 was modified as follow:

[Figure]

Figure 7, why are there spikes on the depth?
The spikes at the convergent wall is due to the lack of boundary fitting of the Cartesian grid. Same boundary effect can be also found in other studies e.g. (Rogers and Fujihara et al., 2001).

Figure 8, is this already with the DBCM? if yes, where is the boundary of the two models in those plots.
Figure 8 is the results of dam-break test case which is used to verify the performance of the hydrodynamic model. Thus, only hydrodynamic model was implemented in this dam-break case.

**References:**

Rogers, B. and M. Fujihara, et al. (2001). "Adaptive Q‑tree Godunov‑type scheme for shallow water equations." International Journal for Numerical Methods in Fluids **35** (3): 247-280.

---

## Author Response (AR1)

**Answers to Anonymous Referee #1 comments**

We would like to thank Referee #1 for the constructive comments, very detail corrections, and recommendations towards improving our manuscript. We are improving our writing quality based on your kind suggestion. These comments are all valuable and very helpful for improving our paper. We appreciate that we have a chance to revise the manuscript as you recommend and to resubmit our manuscript will meet your approval.

In the following, we respond point by point to the comments. The referee comments appear in black and the answers appear in blue.

1/ My first comment is directed to the title. Flood prediction is stated as the main aim. This should be rephrased to flood simulation to avoid confusion. Prediction is often associated with forecast, which is not the aim of this manuscript.

The title has been changed to "A Dynamic Bidirectional Coupled Surface Flow Model for Flood Inundation Simulation" combined with consideration of the second comment.

2/ My second comment is related with the branding of "2D diffusion wave" with "hydrological model". It seems that the authors have developed a 2D diffusive wave model (line 139). If that is the case, this cannot be categorised as hydrological model. The title should be rephrased to "coupled diffusive-full dynamic". Unless the authors can justify the branding of hydrological model, this must be changed.

Some properties of Hydrological models are: a) only propagate information downstream; b) are inherently one-dimensional when simulating flood routing (channels or links), and c) able to simulate other discharge components such as interflow and baseflow besides direct discharge.

Hence it seems that this not fits to the description of the authors. This requires some rethinking and restructuring of the manuscript, but is required in order to avoid misinterpretation of the good work developed.

Indeed, the method proposed in the manuscript was mainly on the coupling of a 2D diffusion wave model and a full-dynamic model. Whereas, the model developed includes the hydrological processes: precipitation, infiltration and runoff routing, so we thought it belongs to a grid based distributed hydrologic model which will include more hydrological processes in the future, such as, evapotranspiration, snowmelt, saturated zones, etc.

Based on this comment, lots of modification and explanation have been made in the modified manuscript. We have modified the title and reorganized the second Methodology section.

3/ My third comment is related to the time steps. how are the time steps being calculated, and how are the two model synchronized?

Time steps are determined by CFL condition as following:

$$\Delta t = C_r \min \left( \frac{\Delta x_i}{|u_i| + \sqrt{gh_i}}, \frac{\Delta y_i}{|v_i| + \sqrt{gh_i}} \right)$$

This formula has added to the end of section of 2.3 Hydrodynamic model in the modified manuscript. In DBCM, the same grid mesh system was used to solve the DWE and SWE. For each grid cell, only one approach whether DWE or SWE is solved except for some special treatment on the coupling boundary. This can be found in the 2.4 Coupling approach section in the modified manuscript.

4/ My forth comment is related with the display of the results. The manuscript is about coupling two different models; however, it is not clear in the plots where the boundary is. Please add to all plots the location of the boundary between the two.

The evolvement of coupling boundary has been added to the V-shape catchment case and Helin Basin case in the modified manuscript. See Figure 14 and Figure 24 in the modified manuscript.

5/ minor comments:

line 60, "evolved" instead of "involved"

Thanks for correction

line 60, "overestimate the flood risk in some extent" is too vague, please rephrase

The text was rephrased in the modified manuscript. See line 51~53 in the modified manuscript.

line 70 to 73, is confusing. e.g. the same sentence starts with "the next step", and ends with "the next time step", is the former not time? What is the meaning of "present flow state"? is that same as current, as stated previously? if yes, always use same wording for the same meaning.

Yes, the former is not time.

"present flow state" means the flow information of each grid cell in current time step.

The text was rephrased in the modified manuscript to make it clear, see line 60~65 in the modified manuscript.

line 80, why is it a "significant problem" please explain.

The problem indicates the bargain of cost and benefit between simulation efficiency and resources as illustrated the following text.

It is explained in line 69~71 in the modified manuscript

line 85, remove "of"

Thanks for correction

line 86, should read "considers"

Thanks for correction

line 91, "apt to", replace with "adequate". to perform what? (not clear). "doesn't" replace with "does not"

Thanks for correction

line 82 should read "further studies are necessary"

Thanks for correction

line 109, what is "slope runoff" - consider removing the word "slope" and use simply runoff throughout the text

Thanks for suggestion.

line 173, is telemac being used, or it was "re-written". please clarify this.

TELEMAC is not used since it is developed based on unstructured grid, while the model in this paper is based on Cartesian grid.

The sentence has been changed into: These equations are solved using the finite volume method similar to TELEMAC (Ata et al., 2013). And the convection flux on grid faces is calculated using the HLL scheme with WAF approach (Toro, 2001).

See line 164~165 in the modified manuscript.

line 193, is the hydrological model 1D or 2D, if it is the latter, it can also produce inundation extents. please add some plots which show the boundary of the "hydrological" and hydraulic model being changed. Overlay these with the flood inundation extent.

The 2D diffusion wave equations were used for the runoff routing process, so it is 2D.

The time varied coupling boundary evolvement has been added to the modified manuscript (Fig.14 and Fig.24).

line 212, "is moved to point A", so where was it before?

This indicates the coupling boundary position at the present time step. For the first case, the coupling boundary position will moved to A in the next step, as shown in the following figure (a).

[Figure]

In the modified manuscript, the paragraph has been rewritten to make it clear. See line 198~204 in the modified manuscript.

lines 215 to end of paragraph is confusing.

The text has been rewritten in the modified manuscript. See line 198~204 in the modified manuscript.

Figures 2 and 4, what is the colour code?

Figure 2 and Figure 4 have been merged into a single figure: Figure 3 in the modified manuscript.

Figure 4, why water cannot flow to the left in the middle figure?

This is because the effects of gravity are that the flow at any point will trend to be in the direction of the steepest water surface slope(Bradbrook et al., 2004), as shown in the following figure:

[Figure]

For clarification, figure 4 (Figure 5 in the modified manuscript) was modified as following:

[Figure]

Figure 7, why are there spikes on the depth?

The spikes at the convergent wall is due to the lack of boundary fitting of the Cartesian grid. Same boundary effect can be also found in other studies e.g. (Rogers and Fujihara et al., 2001).

Figure 8, is this already with the DBCM? if yes, where is the boundary of the two models in those plots.

Figure 8 is the results of dam-break test case which is used to verify the performance of the hydrodynamic model. Thus, only hydrodynamic model was implemented in this dam-break case and no boundary in this case.

**References:**

Bradbrook, K. F., S. N. Lane, S. G. Waller, and P. D. Bates, 2004, Two dimensional diffusion wave modelling of flood inundation using a simplified channel representation: International Journal of River Basin Management, v. 2, p. 211-223.

Rogers, B., M. Fujihara, and A. G. Borthwick, 2001, Adaptive Q‐tree Godunov‐type scheme for shallow water equations: International Journal for Numerical Methods in Fluids, v. 35, p. 247-280.

**Answers to Anonymous Referee #2 comments**

We would like to thank Referee #2 for the constructive comments, very detail corrections, and recommendations towards improving our manuscript. We are improving our writing quality based on your kind suggestion. These comments are all valuable and very helpful for improving our paper. We appreciate that we have a chance to revise the manuscript as you recommend and to resubmit our manuscript will meet your approval. In the following, we respond point by point to the comments. The referee comments appear in black and the answers appear in blue.

The manuscript presents a novel dynamic coupling approach of hydrologic-hydrodynamic simulation, where representation of flood processes and simulation can be potentially improved, and therefore a substantial research topic. However, unfortunately the performance assessment of the model has not been done properly and extensive enough to justify the strength of the proposed coupling scheme. For instance, the performance is only compared to UCMs, instead of BCM, as that's where the novelty value of DBCM lies upon and should be evaluated. Meanwhile, in the comparison to UCMs (section 4), the hydrological consideration using point source inflow boundary is not an appropriate method to support authors' claim (see below point 3).

The key feature of the DBCM, is the coupling boundary where accounting both mass and momentum transfer between hydrologic model and hydrodynamic model. And focus is on the momentum transfer which consider less in existing UCM and BCM. Since UCM always use SWE for simulation, and momentum transfer information can be readily obtained. Thus, in the original manuscript, lots of efforts were put on comparison between UCM and DBCM. Even though, in the modified manuscript, the comparison between UCM, BCM and DBCM has been added to the V-shaped catchment, see Figure 11.

I also think that the presentation of the manuscript can be much further improved in terms of language clarity, and figure presentation.

Based on the comments of the two referees, most sections of the manuscript were rephrased or rewritten. Besides, lots of figures have been regenerated, as can be seen in the modified manuscript.

More specific comments can be found below:

1) In Section 4, Scenarios in table 2 and their related text: It is not sure how the result from the hydraulic model in Case B being transformed into the spatial distribution of water depth, e.g. in figure 20.b. Is the HEC-RAS setup, i.e. in case B, considers both 1D & 2D unsteady flow simulation (i.e. in HEC-RAS v.5) OR just 1D + GIS "bathtub method" OR just 2D? Also please provide at least table or information for these setups in the supplementary so that it benefits others who may want to compare in the future.

Case B use HEC-RAS v.5 2D, and inundation area can be obtained directly through the post process RAS Mapper tool. However, in the modified manuscript, case B has been removed and this issue no longer exists in the modified manuscript.

2) I would also consider improving the presentation of figures 20 & 22, e.g. to remove the information of the elevation in the backgrounds and they may only confuse readers with many colors. Instead, since the authors validated vaguely with the record of water depth reported in the urban areas, the addition of extent, e.g. hollow polygon of these urban areas would be more useful. This

could also point out further the claim of the author regarding case A failing to simulate flood in the urban areas (lines 488 -495), or rather parts of the urban areas (higher elevation).

Figure 20 in the original manuscript has been removed. And regenerated Figure 22 using a satellite imagery base map to make it clear.

3) Line 495-498, with regards to mentioned reasoning of case A and B failing to simulate the flood depth at the higher elevation further from the river bed, I would add the obvious reasoning is the fact that case A and B contains NO distributed hydrologic modelling, they only consider point sources inflow boundary conditions obtained from SWAT, and therefore none of the spatial rainfall distribution is considered. Such failure has NOTHING to do with the lack of dynamic nor bi-directional coupling method. Therefore, the author's approach for the evaluation/ comparison is not appropriate to prove the strength of the author's DBCM.

Indeed, case A and case B in the original manuscript are not appropriate to support the strength of DBCM. In the modified manuscript, case B has been removed. Besides, the evolution of the coupling boundary added. Actually, Helin town case is just an implementation of DBCM.

4) Since the BCM coupling model (i.e. MIKE SHE + 11) is described timely and only compared conceptually along with author's proposed DBCM and UCM, why is the DBCM is only compared with UCM in terms of performance (instead of all 3)?

Results of Mike she add to V-shaped catchment case in the modified manuscript. See Figure 11 in the modified manuscript.

5) The performance of bi-directional against uni-directional coupling for flood modelling has already been compared/accessed and known overtime in literatures for its improved water transfer dynamic representation and result when setup properly. Since the author's novelty for the coupling approach is emphasize on the improved representation of changing extend of boundary or in other words spatial-dynamic on top of the BCM, I would say the strong focus of the finding should be on the performance assessment of author's DBCM over BCM instead of the obvious UCM vs DBCM.

Thanks for your advice. In the modified manuscript, for V-shaped case, we add the results from other researchers, see Figure 11 in the modified manuscript. And remove case B(UCM) in Helin town case.

6) Line 115, : : :involves the "processes of precipitation"? What does it mean within the context?

Actually, we mean "hydrological processes". The sentence has been rephrased, see line 105 in the modified manuscript.

Line 430-440 and Figure 18, not very clear, please provide a better legend for the soil type, what they mean instead of meaningless abbreviation only and imply for the model, e.g. relating to SCS number or others. Also, please consider providing appropriate reference & source to data inputs like GDEMV2, LULC, and soil type database.

Most of the datasets were obtained from online public data center. In data availability section, we have add links to these websites. These can be found in Data availability section.

The legend of the soil type has been updated, see Figure 18b in the modified manuscript.

Line 443, The selected coefficient of rainfall and floods: : : in table 2 & 3, I think you meant Table 3 & 4 instead. Also 'coefficient of flood' is not appropriate.

Thanks for correction. The sentence has been rephrased. See line 422~423 in the modified manuscript.

Line 483, "The red cycle: : :urban area"? Even if it is "circles/ points", I understand they are discharge comparison site (p1, 2 & 3).

Red cycle indicates the urban area of Helin town, not p1, p2 & p3. The figure has regenerated, see Figure 20 in the modified manuscript.

Minor: Figure 17 & 18, the DEM information/ figures are repeated, and besides they are also shown differently/ not consistent.

DEM information in Figure 18 has been removed, see also Figure 18 in the modified manuscript.

Figure 19, I suggest "Simulated Discharge (SWAT)" instead of "Calculated Discharge".

The figure has been updated in the modified manuscript. See Figure 19 in the modified manuscript.

Figure 20 & 22, considers revising the legends (incl. the scale) into one single space since they are the same rather than repeated in 2 separate figures, currently It may give the impression that only elevation legend applies to figure a while depth only to figure b.

Figure 20 has been removed from the manuscript because of the removing of Case B. And Figure 22 has been regenerated using a satellite imagery base map, see Figure 20 in the modified manuscript.

Table 2, "outflow" instead of "out flow".

Thanks for correction

Line 340, "slopes" instead of "lopes".

Thanks for correction

**List of changes of the manuscript**

Based on the comments of the referees, most parts of the sections of the manuscript have been rephrased and rewritten.

The details of the changes made in the manuscript list as following, (line numbers are based on the last manuscript version: nhess-2019-355-manuscript-version1.pdf )

1. The title changed from "A Dynamic Bidirectional Coupled Hydrologic-Hydrodynamic Model for Flood Prediction" into "A Dynamic Bidirectional Coupled Surface Flow Model for Flood Inundation Simulation"

2. Line 11~line22 of the abstract has been rewritten.

3. Line27 "…plays an important role…" changed into " plays a rather important role"

4. Line 29 "With the advances in computation …" changed into "With the advances in numerical computation …"

5. Line 32~line43 has been rephrased

6. Line 44~line51 has been rephrased

7. Line 55 "…can be used easily applied to UCM. " changed into "can be used into UCM simulation"

8. Line 57~line64 has been rephrased

9. Line 67 remove "(Muskingum method, etc) which only consider the precipitation and infiltration processes"

10. Line 70~line75 has been rephrased

11. Line76~line81 has been rephrased

12. Line83~line84 has been rephrased

13. Line85 "The UCM feed hydrodynamic model with the output of hydrologic model as the inflow boundary, and does not assess…" changed into "However, the UCM run the hydrodynamic model using the hydrologic model output as the inflow boundary, and the hydrologic model does not assess…."

14. Line 86 "considers" instead of "consider"

15. Line 87 "Taking the MIKE SHE/MIKE 11" instead of "Taking MIKE SHE/MIKE 11"

16. Line 88 "the velocity of a local grid" instead of "the grid velocity"

17. Line 89 remove of "then"

18. Line 90 remove of "before determining whether the flow impact from hydrologic model is side inflow or outflow"

19. Line 91 "adequate" instead of "apt to", remove of " and perform", "does not" instead of "doesn't"

20. Line 92 remove of "only", "further studies are necessary for general implementation" instead of "further study is necessary to be done for more general implementation"

21. Line 93 "other complicate flow cases" instead of "other more complicate cases"

22. Line 95 "account" instead of "consideration"

23. Line 98 "switching between hydrologic model and hydrodynamic model" instead of "switching of applied hydrologic and hydrodynamic models", "The" instead of "A"

24. Line 99 remove of "a"

25. Line 100 "the proposed DBCM" instead of "DBCM proposed in this paper

26. Line 101 "the flow description of the model is more consistent with…" instead of "the running process of the model is consistent with…"

27. Line 102 "the flow information determination based on the theory of characteristic on the

coupling boundary…" instead of "the flow calculation on the coupling boundary…"

28. Line 104 "and comprehensively consider the local flow…" instead of "comprehensively considering the current flow…"

29. Line 107 "is applied to a real river catchment –Helin town in Chongqing City" instead of "is applied to the Longxi river catchment in Chongqing City"

30. Line 109 "runoff routing" instead of "slope runoff"

31. Line 111 "within" instead of "at", the mass and momentum transfer on the coupling boundary are determined based on the characteristic wave propagation theory which is commonly …" instead of "flow information on the coupling boundary is calculated based on the theory of characteristic wave propagation commonly …"

32. Section 2.1 was divided into two sections: 2.1 Runoff generation and 2.2 Diffusion wave approach, and the following section numbers increase accordingly.

33. Line 114 "The hydrologic model used in this study is a raster-based distributed model." Instead of "The hydrologic model used in this study is a physics, raster-based, and distributed model."

34. Line 115 "involves the hydrological processes, e.g. precipitation and infiltration." instead of "involves the processes of precipitation and infiltration.", "for" instead of "in"

35. Line 116 "The precipitation module reads in record datasets from rainfall stations and rainfall intensity in each grid is interpolated using a spatial interpolation function (Thiessen polygon method, Inverse Distance Weighted, etc.)" instead of "The precipitation module reads in record datasets from a rainfall station and interpolates the data over the whole computational domain using a spatial interpolation function (Thiessen polygon method, Inverse Distance Weighted, etc.)"

36. Line 118 "module" instead of "model"

37. Line 126~line137 rephrased and moved to 2.1 Runoff generation.

38. Line 138 "are" instead of "is"

39. Line 146 "obvious" instead of "significant"

40. Line 151 "at each time step(see Fig.01a)" instead of "at each time step"

41. Line 159 add "The depth definition is show as Fig.1."

42. After line 161, add new figure: Figure 1 typical grid and water depth definition

43. Line 164 "2D shallow water equations are the most widely used hydrodynamic model in inundation simulation (Bradbrook,2006;Yu and Duan, 2014; Yu and Duan, 2017)" instead of "The governing equations for the hydrodynamic model are the widely used 2D shallow water equations."

44. Line 173 "These equations are solved using the finite volume method similar to TELEMAC(Ata et al., 2013)." Instead of "The finite volume method following TELEMAC(Ata et al., 2013) are used to solve these equations."

45. Add the formula of time step after line 189.

46. Line 190 "2.4 Coupling approach" instead of "2.3 Dynamic bidirectional coupling model(DBCM)"

47. Line 191~line 204 has been rephrased

48. Figure 2 has been replaced by a new one, the caption changed accordingly and the explanation (line207~line219) has been rephrased.

49. Line 222~line286: some minor corrections were made. Besides, Figure 4 and Figure 5 were merged into one new figure (Figure 5 in the modified manuscript).

50. Line 292 "and then the performance of DBCM will be verified using a V-shaped catchment." instead of "and then the DBCM will be verified."

51. Line 300 remove of "respectively", "and the oblique angle θ = 8.95°" instead of "and θ = 8.95°"

52. Figure 10 was replaced by a new one and the caption updated.

53. Line 341 "slopes" instead of "lopes"

54. Line 344~line353 make into a new paragraph with some minor corrections.

55. Figure 11 was replaced by a new one. Add several hydrographs from reference models.

56. Line 356~line 357 was rephrased.

57. Line 361~line 370 was removed

58. Line 383~line395 as well as Figure 14 were removed.

59. Add a new figure to show the evolution of the coupling boundary (Figure 14 in the modified manuscript) and the corresponding explanation(line360~365 in the modified manuscript)

60. Add a figure to show the simulation efficiency between SWE, DBCM and DWE (Figure 15 in the modified manuscript) and the corresponding explanation (line 373~line380 in the modified manuscript)

61. Line 405 "DWE" instead of "diffusion wave equation"

62. Line 408 "the momentum transfer need to be taken into consideration in order to get reasonable simulation results" instead of "the application of hydrologic model inevitably leads to errors."

63. The caption of Figure 16 was updated

64. Line 412~line420: some minor corrections were made

65. Line 423~line429 was rephrased

66. Line 430 "The input datasets for DBCM include…" instead of "The model data includes…"

67. Line 435 "…the public data provided by local administrative sectors" instead of "…the public data"

68. Figure 18 was replaced by a new one: the DEM information was removed and the Soil type codes were updated.

69. Line 441~line452 was rephrased

70. Case B was removed and Table 2 was updated accordingly. Case C in the older manuscript was replaced by Case B in the modified manuscript.

71. The legend of Figure 19 was updated: "Simulated Discharge(SWAT)" instead of "Calculated Discharge"

72. Line 461~line475 as well as Figure 20 and Figure 21 were removed

73. Line 480~line498: some minor corrections were made, "Case B" instead of "Case C"
    a) Line 483 "that" instead of "the one in inlet"
    b) Line 486 "following" instead of "follows"
    c) Line 487 remove of "course"
    d) Line 488 "in the catchment" instead of "of the reach"
    e) Line 489 remove of "Reviewing the simulation results(Fig.20), the maximum water depth is generally between 2m and 5m along the banks"
    f) Line 490 "in terms of the" instead of "it comes to"
    g) Line 495 "reason for" instead of "cause of"
    h) Line 496 "due to local" instead of "by"
    i) Line 497 "uphill surface flow" instead of "flow from the uphill slope"

    j)    Line 498 "closer" instead of "approximate"

74. Figure 22 was replaced by a new one using the satellite imagery as the base map and add the red circle to indicate the location of Helin town (Figure 20 in the modified manuscript).

75. The legend of Figure 23: "Case B" instead of "Case C" (Figure 21 in the modified manuscript)

76. Add a new figure to show the evolution of coupling boundary in Helin town (Figure 22 in the modified manuscript), and the corresponding explanation (line471~line474 in the modified manuscript)

77. Line 517~538: the Conclusions section, some minor corrections were made:

    a)    Line 517 "surface flow inundation simulation" instead of "flood prediction and analysis"

    b)    Line 518 "solution scheme" instead of "the solution schemes"

    c)    Line 519 remove of "study"

    d)    Line 520 "determine" instead of "compute"

    e)    Line 524 "alteration" instead of "change"

    f)    Line 525 "accounts" instead of "enables"

    g)    Line 526 remove of "significantly"

    h)    Line 530 "information" instead of "state"

    i)    Line 534 "benchmark tests" instead of "test cases"

    j)    Line 535 "the dynamic switching" instead of "the switching"

    k)    Line 536 "… in simulating surface flow, which hardly…" instead of "…in simulating overland flow, hardly achieved…"

78. Data availability section: line 540~line543, add the source of satellite imagery base map used in Helin town simulation.

[revised manuscript text omitted]

---

## Author Response (AR2)

We would like to thank Referee #1 and Referee#2 for the constructive comments, these comments are valuable and very helpful for improving our paper. We appreciate that we have a chance to revise the manuscript as you recommend and to resubmit our manuscript will meet your approval.

In the following, we respond point by point to the comments. The referee comments appear in black and the answers appear in blue.

**Answers to Anonymous Referee #1 comments**

**Report #2**

Suggestions for revision or reasons for rejection (will be published if the paper is accepted for final publication)

line 380 the sentence on "when implemented in a large catchment..." is in my opinion too much speculative, and should be therefore removed.

Thanks for your suggestion. The sentence has been removed in the latest manuscript.

**Answers to Anonymous Referee #2 comments**

**Report #1**

Dear authors,

I really find your manuscript topic and scientific endeavor for a new type of DBCM scheme worthy of attention and that upon revision, the quality of the manuscript presentation has improved as well.

However, my concern remains that you have not really compared your DBCM scheme with other similar dynamic (D) and bi-directional (B) schemes, and I think that should be the focus of your evaluation and big selling point, instead of inadequately comparing it with other UCM and standalone hydrologic and hydrodynamic models. In the revision process, you responded this with the addition of MIKE SHE only in the simplistic V catchment test. However this MIKE SHE only is just another distributed hydrological model without bidirectional hydrodynamics representation, unless this MIKE SHE is coupled with MIKE 11 hydrodynamic model (which is commonly done) and therefore could represent another DBCM scheme for your proper comparison.

Therefore, I would encourage you to revisit the above mentioned issue for a solid revision to present better and emphasize the selling point of your DBCM coupling scheme in contrast to others.

I wish you best of luck.

We have done a comparison between DBCM and Mike SHE/Mike 11 coupling model. See Section 3.3 in the latest modified manuscript, .discharge processes and water depth profiles were obtained to show the performance of the two coupling models. The results show good performance of the proposed DBCM.

**List of changes of the manuscript**

Based on the comments of the referees, most parts of the sections of the manuscript have been rephrased and rewritten.

The major changes in the manuscript list as following.

(Changes are based on the last manuscript version: nhess-2019-355-manuscript-version2.pdf )

1. Abstract and Introduction section have been rewritten, and lots of minor corrections were made to the other sections.

2. Section 3.3, comparison between DBCM and Mike SHE/Mike11 coupling model was made based on the suggestion of the second referee.

**Compare Results**

| Old File: | | New File: |
|---|---|---|
| **nhess-2019-355-manuscript-version2.pdf** | versus | **nhess-2019-355--2020.09.08-2.pdf** |
| **29 pages (2.58 MB)** | | **32 pages (3.55 MB)** |
| 2020/5/14 22:14:36 | | 2020/9/9 2:46:36 |

**Total Changes**

**673**

**Content**

217 Replacements

290 Insertions

155 Deletions

**Styling and Annotations**

11 Styling

0 Annotations

Go to First Change (page 1)

[revised manuscript text omitted]

---

## Author Response (AR3)

We would like to thank Referee#2 for the constructive comments, these comments are valuable and very helpful for improving our paper. In the following, we respond point by point to the comments. The referee comments appear in black and the answers appear in blue.

**Answers to Anonymous Referee #2 comments**

**Report #1**

Dear authors,

I would like to acknowledge your great response in revising the manuscript, and that you have provided the comparison showcase between your DBCM and the other bidirectional coupling such as MIKE SHE-11 coupled model.

Meanwhile as a minor addition to the conclusion section, is there any notes/ limitations/ challenges for the scientific community to consider if they want to utilize your DBCM coupling schemes? For example, the discussion on computational cost?

In the era of reproducible open science and knowledge for the benefit of scientific community at large, would you be able to include the code access as well, e.g. via your GIT repository?

Nevertheless, I wish you all the best for the publication and the continuation of DBCM research endeavor.

Dear Referee:

Thanks for your agreement with the manuscript revision.

The DBCM developed in present study only accounts for precipitation and infiltration, and lots of other hydrological processes, such as evaporation, groundwater interaction, are under development. Besides, a uniform rainfall station data limits its application to relatively small-scale catchment. Further study involving other physical processes, such as evapotranspiration, interception and snowmelt .etc. as well as distributed rainfall datasets processing will be conducted to improve the proposed model and make it suitable for more general application.

The codes will be committed to GIT repository in the near future, and we'll make it open sourced once the model is optimized. Before that, anyone interesting in our model can contact us with the e-mails provided in the authors' information section of the manuscript. For this issue, we've add an instruction in the Code and Data availability section, and this can be found in the latest modified manuscript.

**List of changes of the manuscript**

Based on the comments of the referees, most parts of the sections of the manuscript have been rephrased and rewritten.

The major changes in the manuscript list as following.

(Changes are based on the last manuscript version: nhess-2019-355-manuscript-version3.pdf )

The conclusions section has been rewritten, and some minor corrections were made in the other parts of the manuscript as follow.

Minor corrections:

1.  Line 19: "It is reported that" instead of "As an example,"
2.  Line 44: "datasets" instead of "data"
3.  Line 49: delete "for the momentum equations"
4.  Line 50: "…are coupled via the balance of water volume…" instead of " …are linked via the calculation of flow discharge.."
5.  Line 54: "existing" instead of "current"
6.  Line 57: add "and momentum"
7.  Line 59: delete "."
8.  Line 60: "coupling boundary" instead of "interfaces"
9.  Line 64: add "of the proposed DBCM"
10. Line 68: "precipitation" instead of "rainfall"
11. Line 75: "while" instead of "and"
12. Line 85: delete ", a different grid is needed for the hydrodynamic models"
13. Line 93: "surface" instead of "level"
14. Line 96: "surface" instead of "level"
15. Line 104: "surface" instead of "level"
16. Line 138: "under the" instead of "by"
17. Line 145: "local" instead of "own"
18. Line 150: "rainfall" instead of "rain"
19. Line 156: "For the case in Fig.3a, the…" instead of "Fig.3a shows the case of the flow on a slope. The…"
20. Line 161: "using the flow information" instead of "by the flow"
21. Line 167: delete "still"
22. Line 202: the caption of Figure 5 was revised
23. Line 205: "high" instead of "small"
24. Line 247: "adjusted" instead of "adjustment"
25. Line 259: "experimental" instead of "measured"
26. Line 310: add "termed by river links"
27. Line 311~315: the sentences were rewritten
28. Line 327: "of the centre channel" instead of "along the channel centerline"
29. Line 343: deleted "The DBCM hydrological model and the hydrodynamic mode are solved simultaneously.", "For the DBCM, the main difference…" instead of "The Main difference…"
30. Line 351~355: sentences were revised
31. Line 365: sentence revised
32. Line 377: "It is about 221km long with a catchment area about 3280km$^2$" instead of "The main

channel length is about 221km and the basin area is about 3180km$^2$"

33. Line 386: "relatively" instead of "open", "and farmlands widely distribute around the main stream," instead of " and the farmland is widely distributed,"

34. Line 394~396: some minor corrections

35. Line 402: delete "the result"

36. Line 407: "However" instead of "Then"

37. Line 409: "by the combination of precipitation process and infiltration" instead of "by the combined action of rainfall and infiltration"

38. Line 422: "obtained in" instead of "from"

39. Line 423: delete "in Case A and Case B"

40. Line 427: Several problems involved in the current river topographic conditions:" instead of "The current river bed conditions are as follows:"

41. Line 429: delete "in this catchment"

42. Line 431: "a flash flood that make the local streets and …" instead of "a flood that local streets and …"

43. Line 443~451: the sentences were re-organized

44. Line 455: "receding" instead of "decreasing"

45. Line 465~487: the conclusions section were re-written, especially for the second paragraph, and a new paragraph was added to the end of this section for the limitations and further study of the present DBCM model.

46. Line 488: "Code and Data availability" instead of "Data availability"

47. Line 489: add "as well as codes"

**Compare Results**

| Old File: | | New File: |
|---|---|---|
| **nhess-2019-355-manuscript-version3.pdf** | versus | **nhess-2019-355--2020.10.29-2.pdf** |
| 32 pages (3.55 MB) | | 31 pages (3.05 MB) |
| 2020/9/9 2:46:36 | | 2020/10/29 23:24:39 |

| Total Changes | Content | | Styling and Annotations | |
|---|---|---|---|---|
| **384** | 161 | Replacements | 9 | Styling |
| | 110 | Insertions | 0 | Annotations |
| | 104 | Deletions | | |

Go to First Change (page 1)

[revised manuscript text omitted]

---

## Author Response (AR4)

**List of changes of the manuscript**

Notes that the latest manuscript is generated by **Latex template** while the previous versions were generated by **MS template**. Thus, type-settings in the current manuscript show large difference from the previous versions, especially for the positions of Figures and tables. Even though, the content of the manuscript are the same with the minor revision version except for some minor corrections list as follow:

(Changes are based on the Minor revision version: nhess-2019-355-manuscript-version4.pdf ):

Minor corrections:
1. Line 80: delete "respectively"
2. Line 213: "key" instead "aim"
3. Line 257: delete "of"
4. Line 341: the caption of Figure 13 updated to "Discharge processes in the centre channel"